# Distributed neural representations of conditioned threat in the human brain

Zhenfu Wen[1,2], Edward F. Pace-Schott[3,4], Sara W. Lazar [3,4], Jörgen Rosén[5], Fredrik Åhs[6], Elizabeth A. Phelps [7], Joseph E. LeDoux [1,8,9,10] & Mohammed R. Milad [1,2,10,11] ✉

Detecting and responding to threat engages several neural nodes including the amygdala, hippocampus, insular cortex, and medial prefrontal cortices. Recent propositions call for the integration of more distributed neural nodes that process sensory and cognitive facets related to threat. Integrative, sensitive, and reproducible distributed neural decoders for the detection and response to threat and safety have yet to be established. We combine functional MRI data across varying threat conditioning and negative affect paradigms from 1465 participants with multivariate pattern analysis to investigate distributed neural representations of threat and safety. The trained decoders sensitively and specifically distinguish between threat and safety cues across multiple datasets. We further show that many neural nodes dynamically shift representations between threat and safety. Our results establish reproducible decoders that integrate neural circuits, merging the well-characterized 'threat circuit' with sensory and cognitive nodes, discriminating threat from safety regardless of experimental designs or data acquisition parameters.

Encountering a threatening stimulus triggers sensory systems to relay multimodal information about the threat. This information is forwarded onto circuits that initiate implicit and explicit defensive actions and form long-term memory representations that the organism utilizes in future encounters[1]. Pavlovian threat conditioning has been a valuable experimental paradigm to study these neural representations[2–6]. Past research within this field has intensively focused on the roles of a few subcortical and cortical structures in how the association between the conditioned and the unconditioned stimuli is formed, and how the defensive responses are generated and subsequently extinguished[3,7–10]. This approach has led to the notion of the so-called 'threat circuit' that mainly includes subregions of the

amygdala, periaqueductal gray, hippocampus, medial prefrontal cortex, and insular cortex. Across species, but especially in the rodent literature, data suggest some specific associations between localized functional activations of these nodes and particular behavioral expressions or processes during the acquisition and extinction of conditioned threat[3,11,12]. For example, the amygdala is important for the expression of conditioned freezing responses and for extinction learning[13]; while the hippocampus is involved in contextual information processing[4].

This simplified model based on Pavlovian reactions does not fully account for the range of complex cognitive and memory representations that are formed, stored, and updated during conditioning and

[1]Department of Psychiatry, New York University Grossman School of Medicine, New York, NY, USA. [2]Faillace Department of Psychiatry and Behavioral Sciences, McGovern Medical School, University of Texas Health Science Center at Houston, Houston, TX, USA. [3]Department of Psychiatry, Massachusetts General Hospital and Harvard Medical School, Charlestown, MA, USA. [4]Athinoula A. Martinos Center for Biomedical Imaging, Massachusetts General Hospital, Charlestown, MA, USA. [5]Department of Clinical Neuroscience, Karolinska Institutet, Stockholm, Sweden. [6]Department of Psychology and Social Work, Mid Sweden University, Östersund, Sweden. [7]Department of Psychology, Harvard University, Cambridge, MA, USA. [8]Center for Neural Science and Department of Psychology, New York University, New York, NY, USA. [9]Department of Child and Adolescent Psychiatry, New York University Grossman School of Medicine, New York, NY, USA. [10]The Neuroscience Institute, New York University Grossman School of Medicine, New York, NY, USA. [11]Nathan Kline Institute for Psychiatric Research, Orangeburg, NY, USA. ✉e-mail: mohammed.r.milad@uth.tmc.edu

extinction learning. While the 'threat circuit' plays an important role in the detection and response to threat, there is a need for studying how this circuit interacts with other cognitive and sensory neural networks to accomplish the complex processes during the encounter of threat[14,15]. Currently, the field lacks established distributed neural representations that are sensitive and reproducible for the encoding of threat and safety cues across stages of threat conditioning and its subsequent extinction learning and memory retrieval. A compelling approach to study this is the machine learning-based multivariate pattern analysis (MVPA)[16]. In contrast with the traditional univariate analysis examining localized activation, MVPA enables the integration of neural patterns across distributed regions to sensitively detect subtle differences between mental processes. The cross-validation and external validation procedures of MVPA provide ways to examine the generalizability of activation patterns. Considerable evidence has shown that the MVPA is more stable and sensitive than traditional univariate analysis[16–18]. In addition, MVPA could examine whether neural representations are similar across conditions or processes even for overlapping univariate activations[19], and thus allows one to test the specificity of observed patterns to conditioned threat compared to other triggers of negative affect.

In this work, we employ MVPA to investigate the neural representations of conditioned threat throughout conditioning, extinction learning, and extinction memory recall using two analytic approaches. In the first approach, we examine fMRI-based neural patterns within the 'threat circuit' in decoding stimuli that have been conditioned to signal threat from those that have been conditioned to signal safety. In the second approach, we examine neural patterns beyond the 'threat circuit' in contributing to conditioned threat and safety representations. Specifically, based on recent conceptualizations that more distributed neural systems are engaged in threat processing[14,15,20–24], we conduct decoding analysis using distributed 'whole-brain' activation patterns (with the 'threat circuit' excluded). We construct decoding models based on a discovery dataset ($n = 425$) from participants that underwent an established two-day threat conditioning and extinction paradigm[25,26], and evaluate the generalizability of the models on two external validation datasets ($n = 220$) using the same paradigm. By combining the 'threat circuit' and the brain regions identified in the second analytic approach, we construct an extended circuit that includes sensory, working memory, and cognitive neural nodes. We then train new classifiers based on neural activations from this extended circuit, and apply these decoders to 7 external datasets (Supplementary Fig. 1). The objective here is to test the sensitivity, specificity, and generalizability of this extended circuit and the newly trained classifier to distinguish threat from safe cues across datasets that used different variants of threat conditioning paradigms as well as other affect-related paradigms. We demonstrate that threat and safety cues can be robustly decoded using activation patterns within and beyond the 'threat circuit'. The neural decoders sensitively and specifically distinguish between threat and safety cues across multiple datasets acquired using varying paradigms. Thus, we establish generalizable and robust decoders for threat and safety in the human brain.

## Results

### Decoding threat and safe cues based on patterns of the 'threat circuit'

First, we used activation patterns of the 'threat circuit', which mainly included basolateral and centromedial amygdala (BLA, CMA), anterior and posterior parts of hippocampus (aHPC, pHPC), subregions of the insular cortex (dorsal anterior part, dAI; ventral anterior part, vAI; posterior part, PI), dorsal anterior cingulate cortex (dACC), subgenual ACC (sgACC), and ventromedial prefrontal cortex (vmPFC), for the decoding analysis. We examined the sensitivity of the 'threat circuit' patterns in distinguishing threat from safe cues using a discovery

dataset ($n = 425$). This discovery dataset was constructed by all available data from our laboratory that used the Milad et al. paradigm[25,26] to maximize the sample size (Supplementary Table 1). The paradigm included three experimental phases. During threat conditioning phase, two cues were paired with a shock (CS + ) and 1 cue was not paired with a shock (CS-). During extinction learning phase, a CS+ and a CS- were presented without shock. The next day, extinction memory recall phase was conducted by presenting all 3 cues: extinguished CS+ (CS + E), unextinguished CS+ (CS + U), and CS-. Considering the dynamic nature of the threat/extinction learning processes[24,27–31], we divided each experimental phase into 4 trial-blocks (TB, from TB1 to TB4). Since each experimental phase contained 32 trials in total, each TB contained 8 trials (4 trials for each CS type). The activation maps of each CS type in each TB were estimated via the general linear model (GLM), and separately used to build a decoding model for each TB. We evaluated the decoding performance (forced-choice accuracy[32–34]) of each trial-block across the two-day experiment using a 5-fold cross-validation procedure.

As shown in Fig. 1A, activations from the 'threat circuit' successfully decoded CS types at all four trial-blocks of threat conditioning (CS+ vs. CS-, TB1-TB4: 70.5%, 74.0%, 65.5%, 68.7%, all $p < 0.001$, permutation test, one-sided, see Supplementary Table 2 for details), the first trial-block of extinction learning (CS+ vs. CS-, TB1: 62.4%, $p < 0.001$; TB2-TB4: 53.5%, $p = 0.077$; 52.9%, $p = 0.12$; 51.2%, $p = 0.26$), and the first trial-block of extinction memory recall (CS + E vs. CS-, 68.0%, $p < 0.001$; CS + U vs. CS-, 67.4%, $p < 0.001$). In the conditioning phase, the shock response was not likely to have a major impact on the classification performance, since we obtained comparable results in decoding unreinforced CS+ from CS- (Supplementary Fig. 2). Considering that the autoregressive model of GLM suffers from removing temporal autocorrelation[35], the preceding shock signal might be picked up by the classifier. To investigate this issue, we conducted an additional analysis exclusively using trials not preceded by a shock and not paired with a shock. The results (Supplementary Fig. 3) demonstrate that the classification performance was not predominantly contributed by shock signal. Next, we tested the generalizability of the decoding models on two external validation datasets ($n = 98$ and $n = 127$, respectively) in which participants underwent the same paradigm as those in the discovery dataset[36,37]. The paradigm setups (e.g., trial number, images used as CS) of the two external datasets were the same as in the discovery dataset. The 'threat circuit'-based decoding models were successfully generalized to the two validation datasets (Fig. 1B, C, see Supplementary Table 3 for details).

We next examined the voxel contributions to the successful decoding to obtain a more refined spatial resolution of the CS representations within these nodes (Fig. 1D). The classifier weights were transformed into predictive patterns using Haufe transformation to provide a robust interpretation of voxel contributions[38]. Compared to classifier weights, the predictive patterns are often smoother and better resemble mass-univariate analysis[38]: voxels with positive predictive patterns (shown in red in Fig. 1D) show more activation to CS+ presentation, while voxels with negative predictive patterns (shown in blue in Fig. 1D) show more activation to CS- presentation. This analysis showed several interesting results. First, subregions of the amygdala and insular cortex differentially contributed to decoding the CS+ and CS- across trial-blocks. For example, during early conditioning and recall, the BLA mostly exhibited predictive patterns towards the CS-, while the CMA exhibited predictive patterns associated with the presentation of the CS + , only during the first trial-block of conditioning, which then subsequently switched to encoding for the CS-. Second, several regions exhibited consistent predictive patterns across decoding models of different trial-blocks. Specifically, dACC and dAI consistently coded CS + , while the hippocampus and PI mainly coded CS-. Third, the predictive patterns of regions such as amygdala, sgACC, and vmPFC were adaptive- changing the coding signal between the CS+

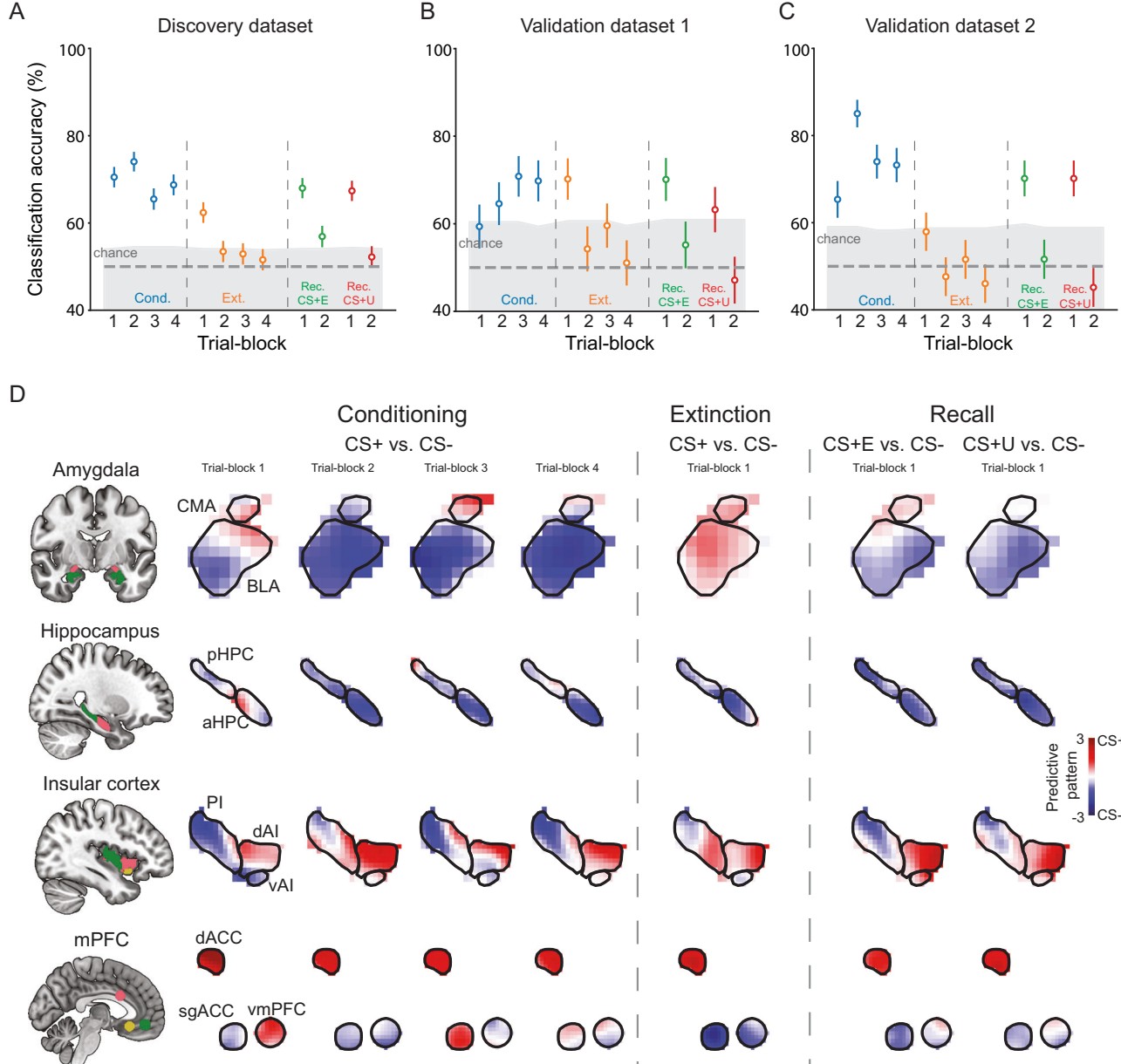

**Fig. 1 | Classification analyses based on neural activations of the 'threat circuit'.** **A** Cross-validation performance on the discovery dataset ($n = 425$). We divided each experiment phase into 4 trial-blocks, and separately did cross-validation on each trial-block. The vertical dash lines separate trial-blocks of different phases. The accuracies are based on forced-choice test. The top of the gray-shaded area indicates the threshold of significant accuracy based on the permutation test ($p < 0.05$, one-sided). **B**, **C** Generalization performance on the validation dataset 1 ($n = 98$) and the validation dataset 2 ($n = 127$). The classifiers were trained based on the discovery dataset, and directly applied to the validation datasets. Error bars in (**A**–**C**) are SEM for accuracies. **D** Predictive patterns of the nodes of the 'threat circuit'. A voxel with red/blue color indicates that this voxel is more activated to CS +/CS-. For the extinction learning and extinction memory recall phase, only predictive patterns of the first trial-block were shown because the other trial-blocks did not show robust classification performance. For the memory recall phase, CS + E vs. CS- and CS + U vs. CS- were shown separately. Cond. Conditioning, Ext. Extinction learning, Rec. Extinction memory recall.

and CS- across trial-blocks and phases, suggesting that these regions were dynamically involved across the experiment. It is important to note that we did not conduct significance tests on the changes of encoding across trial-blocks within each of these subregions, and that these changes may reflect those similar to a habituation effect.

### Decode threat and safe cues based on patterns of the whole brain

The decoding performances based on the 'threat circuit' activation patterns are encouraging, and support the well-documented contributions of these neural nodes to associative learning during threat conditioning and extinction learning. Yet as noted in the introduction, there are several prior studies now pointing out the significant contributions of multiple other brain regions to threat and safety signaling. We therefore moved forward to examine how activations from multiple sensory, attention, and cognitive circuits might also contribute to the encoding of CS+ and CS- information across learning phases. Indeed, using 'whole-brain' activation patterns (excluding the 'threat circuit') led to numerically superior decoding performances (Fig. 2A). We obtained decoding accuracies above 70% at all trial-blocks of threat conditioning (TB1-TB4: 88.6%, 85.7%, 81.7%, 77.5%, all $p < 0.001$, permutation test, one-sided, see Supplementary Table 2 for

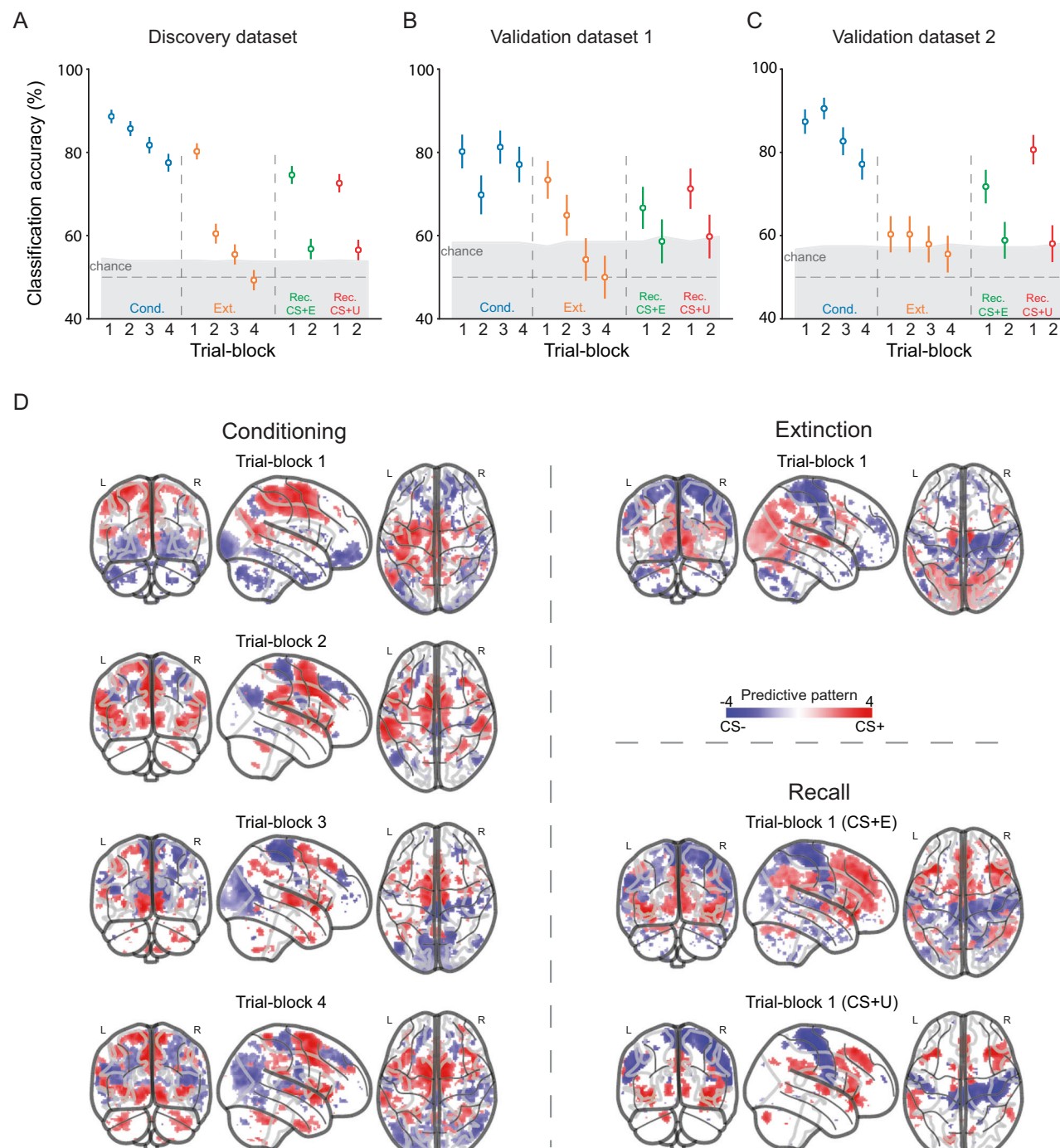

**Fig. 2 | Classification analyses based on neural activations beyond the 'threat circuit'. A** Cross-validation performance on the discovery dataset (*n* = 425). Voxel activations from the whole brain (with voxels that belong in or are close to the 'threat circuit' excluded) were used for the classification. **B, C** Generalizations on the validation dataset 1 (*n* = 98) and the validation dataset 2 (*n* = 127). Error bars in panels A–C are SEM for accuracies. **D** Predictive patterns of voxels across the brain. A voxel with red/blue color indicates that this voxel is more activated to CS + /CS-. Permutation tests were conducted to assess the voxel contributions to the classification. Only voxels that significantly contributed to the classification (*p* < 0.05, FDR-corrected, two-sided) are shown.

details), early extinction learning (TB1: 80.2%, *p* < 0.001), and early extinction memory recall (CS + E vs. CS-, TB1: 74.6%, *p* < 0.001; CS + U vs. CS-, TB1: 72.6%, *p* < 0.001). The decoding accuracies on the external datasets were comparable to those obtained on the discovery dataset (Fig. 2B, C, see Supplementary Table 3 for details), suggesting good generalizability of the decoding models. The numerically superior classification performance of the 'whole-brain' pattern over the 'threat circuit' remained when the number of voxels was matched

(Supplementary Fig. 4). For completeness, we conducted additional analysis by using all gray matter voxels (including the 'threat circuit' voxels) for the classification, which resulted in similar performance as those in Fig. 2A (Supplementary Fig. 5).

We next used permutation tests to define voxel contributions to the classifier across the brain. These analyses identified distributed brain regions that extended into multiple neural systems (Fig. 2D, and Supplementary Fig. 6). For extinction learning and extinction memory

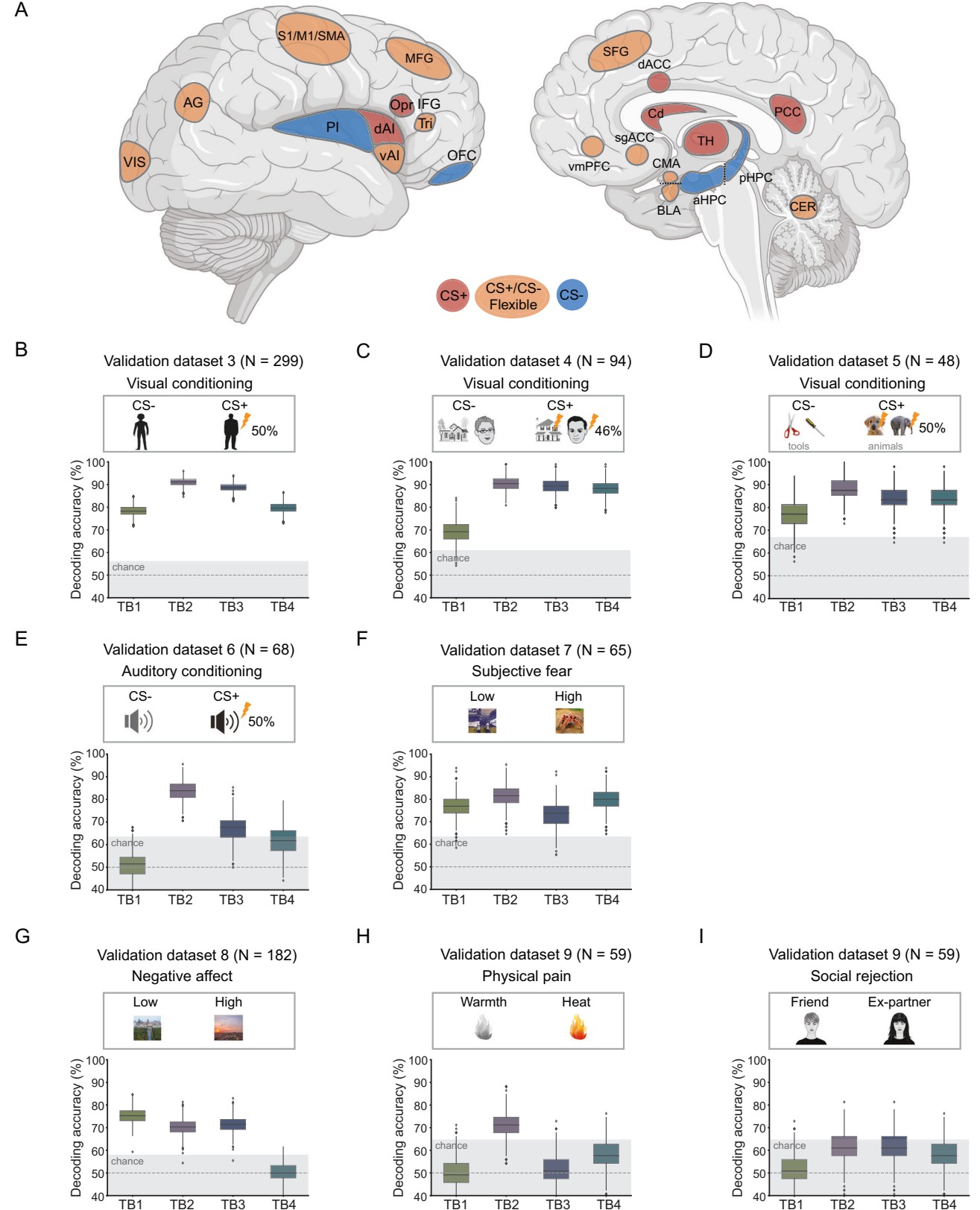

recall phases, we only show the predictive patterns of the first trial-blocks because these were the trial-blocks that showed the most robust classification performance in the two phases. Based on these predictive patterns, we identified 14 representative brain regions that significantly contributed to the decoding across multiple trial-blocks (Fig. 3A), including: (1) angular gyrus (AG), (2) orbital frontal cortex (OFC), (3) supplementary motor area (SMA), (4) primary somatosensory cortex (S1), (5) primary motor cortex (M1), (6) visual cortex (VIS)/Occipital pole, (7) cerebellum (CER)/Crus I, (8) thalamus (TH), (9) opercular part of the inferior frontal cortex (Opr), (10) triangular part of inferior frontal gyrus (Tri), (11) middle frontal gyrus (MFG), (12) caudate nucleus (Cd), (13) superior frontal gyrus (SFG), (14) posterior

**Fig. 3 | The 'threat detection and flexible responding circuit' and its generalizations to external datasets. A** Schematic illustrative map of the extended 'threat detection and flexible responding circuit'. Red-colored regions consistently code the CS+ across experimental phases. Blue-colored regions consistently code the CS- across experimental phases. Yellow-colored regions dynamically code CS+ or CS- depending on the experimental phase. **B** Performances on Vinberg et al. dataset[44] (n = 299). The 4 decoding models (for trial-blocks 1–4; TB1-TB4) from the conditioning phase were applied to the external datasets separately. **C** Performances on the Visser et al. dataset[45] (n = 94). **D** Performances on the Hennings et al. dataset[43] (n = 48). **E** Performances on the Reddan et al. dataset[46] (n = 68). **F** Performances on the Zhou et al. dataset[22] (n = 65). **G** Performances on the Chang et al. dataset[34] (n = 182). **H** Performances on the Woo et al. dataset[19] that used a physical pain paradigm (n = 59). **I** Performances on the Woo et al. dataset[19] that used a social rejection paradigm (n = 59). Bounds of the box represent the 1st (25%) and 3rd (75%) quartiles, the central line represents the median, the whiskers represent the values within 1.5 times of the interquartile range, the flier points represent outliers falling beyond the whiskers. Images in (**A**) were created with BioRender.com. The displayed examples of experimental pictures in (**B**-**I**) are royalty-free pictures obtained from pixabay.com. These pictures are displayed for illustration purposes and are not the pictures included in the original stimulus sets. AG angular gyrus, OFC orbital frontal cortex, SMA supplementary motor area, S1 primary somatosensory cortex, M1 primary motor cortex, VIS visual cortex, CER cerebellum, TH thalamus, IFG inferior frontal gyrus, Opr opercular part of the IFG, Tri triangular part of the IFG; MFG, middle frontal gyrus; Cd, caudate nucleus; SFG, superior frontal gyrus; PCC posterior cingulate cortex, vmPFC ventromedial prefrontal cortex, dAI dorsal anterior insular cortex, vAI ventral anterior insular cortex, PI posterior insular cortex, dACC dorsal anterior cingulate cortex, sgACC subgenual anterior cingulate cortex, aHPC anterior hippocampus, pHPC posterior hippocampus, BLA basolateral amygdala, CMA centromedial amygdala.

cingulate cortex (PCC). The results of identified regions were robust to criteria in defining the percentage of voxels included for the analysis (Methods, Supplementary Fig. 7). Although the OFC was not among the regions that made the largest contributions, we included it because it significantly contributed in the first trial-block of conditioning, which is consistent with literature suggesting its important role in threat-related processing[39,40]. These regions spatially overlap with neural systems including the somatomotor network (SMN), ventral attention network (VAN), frontoparietal control network (CON), and default mode network (DMN). Although the precise functional specializations of these neural systems are unknown in the current task, they have been extensively associated with many high-level aspects of human cognition, including attentional processes, cognitive control, and conscious awareness[41].

## An extended circuit for threat processing
Based on the regions that exhibited significant contribution to the 'whole-brain' decoding models, and the traditional 'threat circuit', we constructed an 'extended threat detection and responding circuit' (Fig. 3A), which, based on our data, better defines the brain regions involved in threat processing than the 'threat circuit' alone. We used representational similarity analysis[42] to test the contribution of each node within this updated circuit to the decoding of threat and safety (Supplementary Fig. 8). Based on each region's preferences of the CS type across trial-blocks, we identified three region communities. The first community (red color in Fig. 3A) consistently coded for threat (CS+) across conditioning, extinction learning, and recall, which included the dACC, dAI, PCC, Opr, Cd, and TH. The second community (blue color in Fig. 3A) consistently coded the safety of the cue (CS-), which included the PI, aHPC, pHPC, and OFC. The third community (yellow color in Fig. 3A) showed dynamic coding; shifting their signal between threat and safety (CS+ or CS-) depending on the experimental phase. This community included the BLA, CMA, vAI, vmPFC, sgACC, AG, S1, M1, SMA, MFG, VIS, Tri, and CER. Overall, these results suggest that the first two communities are likely tracking the valence of the CS cues as negative and positive, respectively, while the third community represents circuits needed to change how we perceive and regulate our responses to the conditioned threat cues. These dynamic brain regions may underlie adaptive learning and flexible responses to salient cues observed throughout the different phases of the threat conditioning protocol.

## The sensitivity and specificity of the extended circuit in decoding threat stimuli
Lastly, we investigated how sensitive and specific the extended circuit decoders are when classifying threat-conditioned stimuli across different paradigms. To accomplish this, decoding models were trained anew by using the activation patterns of the extended circuit only on our discovery dataset, and then separately applied to each of the 7 external datasets with different experimental paradigms (summarized in Supplementary Table 4), including variant paradigms of Pavlovian conditioning, paradigm assessing 'subjective fear' to frightening images, and paradigms of negative affect to intrinsically salient stimuli. We expected that the decoding accuracies on threat, especially conditioned-threat stimuli would be relatively higher than on the other intrinsically salient stimuli. Given that most of the external datasets contain only threat conditioning data, we focused the analysis on the 4 decoding models from the conditioning phase.

We first tested the extended models on three visual threat conditioning datasets (n = 299, n = 94, and n = 48, respectively) that differed in visual stimuli, trial structure, and experimental durations[43–45]. All models showed significant accuracies in classifying CS+ from CS- across all three independent datasets (Fig. 3B–D): the accuracies for the three datasets were: dataset 3 (TB1-TB4: 78.3%, 91.0%, 88.6%, 79.6%, all $p < 0.001$; two-sided binomial test, see Supplementary Table 5 for details), dataset 4 (TB1-TB4: 69.1%, 90.9%, 89.4%, 88.3%, all $p < 0.001$), and dataset 5 (TB1-TB4: 79.2%, 87.5%, 85.4%, 83.3%, all $p < 0.001$). We next tested the decoding models on a dataset (n = 68) that used an auditory threat conditioning paradigm[46], which again yielded significant accuracies in 2 of the 4 models (TB1-TB4: 51.4%, $p = 0.90$; 83.8%, $p < 0.001$; 67.6%, $p = 0.005$; 61.7%, $p = 0.068$; Fig. 3E). We next applied the models to another dataset (n = 65) that assessed 'subjective fear'[22] induced by frightening images, which also revealed significant accuracies (TB1-TB4: 76.9%, 81.5%, 73.8%, 80.0%, all $p < 0.001$; Fig. 3F). We then tested the specificity of the models by applying them to other paradigms that tested neural responses to intrinsically salient stimuli. The accuracies of the classifiers were numerically lower when decoding neural representations about picture-evoked negative affect[34] (n = 182, high vs. low emotionality, TB1-TB4: 75.3%, $p < 0.001$; 70.3%, $p < 0.001$; 72.0%, $p < 0.001$; 50.5%, $p = 0.94$; Fig. 3G). On another dataset (n = 59) that examined neural responses to pain and rejection[19], most of the models showed numerically reduced accuracies for physical pain (high vs. low pain, TB1-TB4: 49.2%, $p = -1.0$; 71.2%, $p = 0.002$; 50.8%, $p = -1.0$; 57.6%, $p = 0.30$; Fig. 3H) or social rejection (friend vs. ex-partner, TB1-TB4: 50.8%, $p = -1.0$; 61.0%, $p = 0.12$; 61.0%, $p = 0.12$; 57.6%, $p = 0.30$; Fig. 3I). For each trial-block, we further compared the decoding accuracies across paradigm types (see Methods). Decoding accuracies for conditioned threat (datasets 3–6) were not significantly different from those for 'subjective fear' (dataset 7) (Chi-square test, TB1: $\chi^2(1) = 0.26$, $p = 0.81$, odds ratio [OR] = 0.81, 95% confidence interval [CI] = [0.41, 1.53]; TB2: $\chi^2(1) = 3.14$, $p = 0.15$, OR = 1.98, 95% CI = [0.90, 4.07]; TB3: $\chi^2(1) = 5.22$, $p = 0.09$, OR = 2.11, 95% CI = [1.08, 3.99]; TB4: $\chi^2(1) = -0.0$, $p = -1.0$, OR = 0.95, 95% CI = [0.46, 1.85]; FDR-corrected, two-sided), and significantly higher than those for intrinsically salient stimuli (datasets 8–9) at trial-blocks 2–4 (TB1: $\chi^2(1) = 1.05$, $p = 0.31$, OR = 1.21, 95% CI = [0.85, 1.72]; TB2: $\chi^2(1) = 47.49$, $p = 5.5E-12$, OR = 3.88, 95% CI = [2.56, 5.90]; TB3: $\chi^2(1) = 30.60$, $p = 3.2E-8$, OR = 2.80, 95% CI = [1.91, 4.11]; TB4: $\chi^2(1) = 55.63$, $p = 8.8E-14$, OR = 3.46, 95%

CI = [2.46, 4.90]; FDR-corrected, two-sided). These data suggest that the neural representations for conditioned threat are distinct from those evoked by intrinsically salient stimuli.

## Discussion

By examining neuroimaging data obtained across multiple experimental paradigms from 1465 participants, we show that brain responses to stimuli representing conditioned threat, its subsequent extinction learning, and memory recall can be classified using activation patterns from the 'threat circuit'. Importantly, however, the classification accuracies were numerically superior when activation patterns from other distributed neural systems (excluding the 'threat circuit') were used. Based on these findings, we constructed the 'extended threat detection and responding circuit' and validated its sensitivity and specificity in classifying conditioned threat. The trained classifiers are sensitive and reproducible regardless of experimental paradigms or MRI scanners across countries. Our results continue to support the important role of the 'threat circuit' in the detection and response to threat. These results support current conceptions in the field by showing that sensory and cognitive neural nodes are also essential in enhancing the classification accuracies related to threat and safety processing; thereby highlighting the importance of considering multiple neural systems in concert for a comprehensive understanding of the underlying mechanism of threat encounters.

A few studies have employed MVPA methods to examine neural representations during threat conditioning and extinction[43,46–48]. The datasets used in these studies, however, were much smaller ($n < 70$) and contained one version of the conditioning paradigms, thus raising concerns regarding the sensitivity and generalizability of MVPA studies[49,50]. Unlike prior studies, our results are based on a much larger sample size from multiple datasets ascertained across different experimental paradigms. Using activation patterns within the 'threat circuit', we decoded threat from safety cues across experimental phases of threat conditioning, extinction learning, and extinction memory recall (Fig. 1A), which showed robust generalizability to two validation datasets (Fig. 1B, C). These findings are consistent with ample evidence suggesting the important role of the 'threat circuit' in threat detection and responding[3,7,9,21]. Many rodent studies showed that neural manipulations within the amygdala and parts of the ventromedial prefrontal cortices significantly modulated conditioned responses (for review, see refs. [11,12,51]). Human lesion studies also provide valuable support to the necessary contributions of neural nodes like the amygdala and medial prefrontal cortex to conditioned threat and negative affect processing[52–57]. Due to challenges related to the spatial resolution of our fMRI data, we did not include regions like periaqueductal gray and bed nucleus of the stria terminalis (BNST) into the 'threat circuit', although previous studies showed their important roles in fear and/or anxiety-related processing[11,58].

In addition to the 'threat circuit', our data highlight the value of other nodes that are also important in conditioned threat processing. Importantly, numerically higher decoding performances were obtained when including neural patterns from other systems that go beyond the 'threat circuit'. When we encounter a threat, we need to process all sensory information, pay attention to our surroundings, retrieve prior memories related to the encounter of this or similar threats, and decide how to respond. Indeed, our results do suggest an engagement of multiple sensory and cognitive systems during the encounter of threat and safety cues, which is likely needed for the processing of these complex representations. Our results provide compelling empirical evidence for conceptual frameworks that threat processing involves multiple processes encompassing multiple brain circuits[14,21]. We identified 14 representative regions spanning across the primary sensory and motor cortex, frontal-parietal cortex, and cerebellum, that significantly contributed to the decoding models across threat conditioning and extinction learning phases. The high

decoding performances here suggest the important role of these regions in supporting implicit and explicit representations of threat and safety. While the OFC only significantly contributed to the early conditioning phase, we included it because it was reported to be involved in threat-related processing across studies[39,40]. There might be other regions that are only transiently engaged at specific stages of conditioning and/or extinction (like the OFC here), that were not identified. Our objective here is not to thoroughly identify all the brain regions that are involved in the processing. Rather, we are highlighting a more integrative view in aggregating the 'threat circuit' together with other neural systems involved in sensory and cognitive processes, to study neural mechanisms underlying threat encounter, as proposed in recent conceptualizations[14,15,20–24].

Our analyses of voxel contributions to the decoding of CS+ and CS- across experimental phases revealed interesting predictive patterns (Figs. 1D, 3A). Specifically, parts of the insular cortex (posterior), medial OFC, and parts of the hippocampus, continuously encode the representation of the safe stimulus (CS-) regardless of the experimental phase. In contrast, multiple brain regions, e.g., the dAI and the dACC consistently contributed to the classification of the CS+, suggesting a role in the encoding of the associative aspect related to threatening cues. These brain regions continued to signal, or encode, the CS+ even during extinction learning and during extinction memory retrieval, consistent with maintaining some representation of the initial conditioned threat memory regardless of extinction[3]. These data provide neurobiological support from the human brain to the now well-accepted concept that extinction does not erase the original conditioned threat associations, first proposed by Pavlov in 1927[59]. Of note, the brain regions discussed above also respond to a broad range of stimuli[60–63], and our experimental design and analyses used in our study could not determine whether they are preferentially associated with Pavlovian threats vs. other processes. Several neural nodes showed dynamic preferences in the encoding of the CS+ and CS- across experimental phases, especially those associated with sensory processing, attention, and cognitive control like the MFG, SFG, vmPFC, and amygdala, amongst others (results based on the representational similarity analysis, Fig. 3A and Supplementary Fig. 8). For example, sub-nuclei of the amygdala (BLA and CMA) exhibited preferences to CS+ during early extinction learning, while BLA exhibited preferences to CS- and CMA exhibited preferences to CS+ types during early threat conditioning and extinction memory recall (Fig. 1D). This observation is consistent with rodent and human studies showing that amygdala contributes to threat and safety processing in an anatomically specific way[30,64–66]. We refer to those neural nodes that dynamically code CS+/CS- as 'flexible' coders.

Why do the 'flexible' coders change their encoding of the CS type? One possible explanation for this change is that it may reflect training-induced dynamic change in neural representation- similar to a habituation effect. Another plausible explanation might be related to the fast neural encoding during associative learning and/or rapid behavioral flexibility that accompanies this learning. During threat conditioning and extinction learning, the significance of the conditioned stimuli changes across the learning phases, which requires a dynamic learning signal. Associations between each CS color and the presence or absence of shock occur in the early trials of threat conditioning. Similarly, during early extinction, the distinction between the CS+ which represents threat, and the CS- which represents safety, is diminished due to extinction learning within a few trials. The diminished difference between CS+ and CS- led to rapidly decreased decoding accuracy after early extinction which ended up in chance-level performances in late extinction (Figs. 1A–C, 2A–C). The same pattern of drop in accuracies was also observed during extinction memory recall. This is the case for the extinguished (CS+E) and unextinguished (CS+U) cues. The rapid decrease in accuracies is consistent with the well-documented exceptionally quick extinction learning in humans both during extinction

learning and extinction memory recall[3,67]. During these learning phases, the subjective feelings, behaviors, and psychophysiological indices are also dynamically changing. These dynamic neural coders could therefore underlie learning/behavioral changes at varying times during each experimental phase. In support of this idea, we recently showed that activations within sub-nuclei of the amygdala change during threat conditioning in a temporally specific manner[30]. Temporal dynamics across distributed neural networks in activation and functional connectivity also change trial-by-trial during extinction learning, suggesting extinction-induced neural plasticity[24,68]. Future studies focusing on the temporal dynamics of these brain activations and how they might be linked to the encoding of neural plasticity or behavioral outcomes are needed to further clarify the flexibility of these nodes to encode threat and safety cues.

The external threat conditioning datasets used for validations were acquired using different MRI scanners and experimental paradigms that were substantially distinct from the Milad et al. paradigm[25,26] used to train the classifier. These paradigms used a variety of conditioned stimuli, and applied complex and differing reinforcement schedules. Despite all of these variations, the models trained on one paradigm (the Milad et al. paradigm) were still sensitive and specific in distinguishing threatening from non-threatening conditioned cues across paradigms (Fig. 3B–D), suggesting that the classifiers learned some general patterns of conditioned threat. A note of caution to make here is that our trained models cannot make a distinction between decoding features that represent the general processing of conditioned threat from features that might be specific to the paradigm used to train the classifiers (such as specific colors or modality of CS, reinforcement schedule, etc.). The classification accuracies were numerically lower when our trained classifiers were applied to a study that used auditory fear conditioning (Fig. 3E), supporting the idea that some coded information in the original classifiers was indeed specific to the paradigm on which it was trained. As expected, the accuracies of our models tested in the negative affect and pain processing studies (Fig. 3G–I) were significantly lower (based on the Chi-square test) compared to those obtained from the threat conditioning studies (Fig. 3B–E). Despite the low accuracies, some of these models were still significant, suggesting a partial overlap in neural coding between conditioned threat and those related to general negative affect. Additional studies are needed to further parse out aspects of the neural representations that are specific to the paradigm used to train the models from those that are unique to threat and negative affect processing.

Several limitations of this study should be noted. First, the masks for some 'threat circuit' regions, including vmPFC, sgACC, and dACC, were defined using small spheres around coordinates derived by meta-analysis. These masks could not fully cover the three relatively large cingulate cortices. Thus, patterns of activations within voxels outside of these masks may have contributed to the 'whole-brain' classification. We note that the size of the mask definition is unlikely to have impacted the 'whole-brain' classification results. This is because we conducted additional analyses using much larger masks (thus excluding more voxels from the same cingulate regions) and the outcome of the 'whole-brain' classifiers did not change (Supplementary Fig. 9). Second, the regional coding preference pattern summarized in Fig. 3A (consistent for CS + , CS-, or flexible) represents a general schematic illustration of the activation patterns specific to our experimental design and its structure. We speculate that while the engagement of these brain regions across different experimental stimuli and experimental design might be consistent, their decoding patterns across experimental phases might differ from one study to another. Further studies are needed to examine the dynamic neural representations of these regions across paradigms. Lastly, the results obtained from this study should not be misinterpreted to infer causality, given the indirect nature of fMRI data and the lack of direct manipulations of the brain signals. Future mechanistic studies are needed to test how the patterns of activations we observed might influence, or lead to, behavioral changes pertaining to the expression and extinction of threat responses across experimental phases.

In summary, our results point to the important contribution of multiple sensory and cognitive neural nodes to the decoding of stimuli associated with threat detection and responding. While prior studies have indeed reported the engagement of sensory and cognitive brain regions during threat conditioning and extinction, our study provides evidence of an integrative circuit that encodes for threat and safety, consistently, specifically, and sensitively- irrespective of experimental paradigm variations or site of data acquisition. Our results support recent conceptualizations calling for the integration of broad networks required for the implicit and explicit processing of threat. Cortical nodes involved in attention and sensory perception in other animal models have received relatively less attention when examining threat conditioning and extinction. We propose that broadening our focus to multiple cortical nodes would provide valuable data to further our understanding of how we detect and respond to threat. A potential application for the decoders is in neuromodulation studies where we could envision inducing some neuromodulation on one or more of the nodes from the extended circuit and evaluate how such manipulation might change the decoding accuracies and the dynamic interactions and encodings we observed.

## Methods

### Participants

We analyzed data from a total of 1465 participants across multiple studies. No statistical method was used to predetermine sample size. No data were excluded from the analyses. Participants from the discovery dataset ($n = 425$) were combined from multiple previous studies[25,26,69–73] and unpublished studies from the Milad lab (see Supplementary Table 1 for details). The methods used to ascertain the new unpublished data, including all experimental procedures conducted were identical to those used for all of our published studies. The data of the external validation datasets 1 ($n = 98$) and 2 ($n = 127$) were from two other studies[36,37]. All participants in those datasets underwent the Milad threat conditioning and extinction paradigm[25,26]. Data from the external validation datasets 3–9 were from other studies that used variants of threat conditioning paradigms or other affect-related paradigms[19,22,34,43–46] (see Supplementary Table 4 for the summary). As stated in all of the above-referenced studies, all procedures were approved by the Institutional Review Boards of the corresponding research sites. The procedures for collecting the new unpublished data were approved by the Institutional Review Board of the Massachusetts General Hospital. All participants provided written informed consent before they participated in the studies.

### Experimental procedure

The description of the experimental procedures and the neuroimaging data acquisition section below pertain to only the participants from the discovery dataset and the validation datasets 1–2. Descriptions of experimental procedures and statistics for validation datasets 3–9 will be provided below. Participants underwent a validated 2-day fear conditioning and extinction paradigm[25,26] during fMRI scanning. On day 1, participants were first exposed to threat conditioning, during which they were presented with three colored lights (conditioned stimuli, CS). Two of the lights were paired with a mild electric shock (CS + , 8 trials for each CS + , reinforcement rate: 62.5%) and the other was not (CS-, 16 trials). Threat conditioning was followed by an extinction learning phase, when the CS- and one of the CS+ were repeatedly presented in the absence of shock (16 trials for each CS). On day 2, participants underwent a memory recall phase to assess their extinction memory. During this phase, all three lights: the extinguished CS+ (CS + E, 8 trials), the unextinguished CS+ (CS + U, 8 trials), and the

CS- (16 trials) were delivered without shock. The trial structure was similar across experimental phases. Specifically, each trial started by presenting a context image (either a library or an office) for 3 s, after which the CS was presented for 6 s. In a reinforced CS+ trial, a shock was delivered at the end of the CS for 500 ms. In other CS trials, no shock was delivered. A fixation was then presented during the inter-trial intervals, which ranged between 12 and 18 s, with an average of 15 s. The order of trials was pseudorandom. The colored lights used as CS+s and CS- were counterbalanced across participants. The conditioning occurred in one context (e.g., the 'office' image) while the extinction learning and extinction memory recall phases occurred in the other context (e.g., the 'library' image). The context assignments were counterbalanced across participants.

## Neuroimaging data acquisition and preprocessing
Neuroimaging data from the discovery dataset and the validation datasets 1–2 were acquired using three different MRI settings. Details about the different settings are listed in Supplementary Table 1. Within the discovery dataset, data from 121 participants were collected with setting 1[25,26], and data from 304 participants were collected with setting 2[69,70]. Data from validation dataset 1 were collected with setting 3[36], and data from validation dataset 2 were collected with setting 2[37].

Imaging data were preprocessed using fMRIPrep 20.0.2[74]. The T1-weighted (T1w) were corrected for intensity non-uniformity with N4BiasFieldCorrection (ANTs 2.3.3) and used as T1w-reference throughout the preprocessing. The T1w-reference was skull-stripped, segmented into cerebrospinal fluid, white-matter and gray-matter, and then spatially normalized into the Montreal Neurological Institute (MNI) space (MNI152NLin2009cAsym) through nonlinear registration with antsRegistration (ANTs 2.3.3). The functional images were head-motion corrected using mcflirt (FSL) and slice-timing corrected using 3dTshift (AFNI). The preprocessed functional images were then co-registered to the T1w-reference using flirt (FSL) with the boundary-based registration (nine degrees of freedom), and spatially normalized into the MNI152NLin2009cAsym space by applying the parameters obtained from T1w-reference spatial normalization. Normalized functional images were resampled to $2 \times 2 \times 2$ mm voxel size using Lanczos interpolation (ANTs 2.3.3) and smoothed with a 6-mm full-width half-maximum (FWHM) Gaussian kernel.

## Brain activation estimation
We used a univariate general linear model (GLM) implemented in the Nistats 0.0.1rc toolbox to estimate activation maps of different stimuli. Since previous human and rodent studies have shown distinct neural activations across trials even within an experimental phase[24,27–30], we divided trials of each CS type into trial-blocks (4 trials per trial-block) and separately modeled the brain activity to each trial-block. For the conditioning or extinction phase, there were 4 trial-blocks for each CS. For the recall phase, there were 2 trial-blocks of CS + E/CS + U and their corresponding CS- trial-blocks. For each trial-block, the regressor was modeled as boxcar functions time-locked to the CS presentations and convolved with the canonical hemodynamic response function (HRF). We also included the 6 motion parameters, high-pass temporal filtering (128 s) terms, volume-censoring indicators (volumes with framewise displacement >0.9 mm[75]), and polynomial drift in the GLM model as nuisance variables. A first-order autoregressive model was used to account for the temporal structure of the noise. For each participant, the GLM model resulted in two activation maps for each trial-block (one for CS + , one for CS-), which were used for the following decoding analysis.

## The first analytic approach: decoding based on the 'threat network' patterns
We separately trained classifiers to discriminate CS type at each trial-block (conditioning: CS+ vs. CS-, extinction: CS+ vs. CS-, recall: CS + E vs. CS- and CS + U vs. CS-) based on the estimated brain activations. Voxel activations from the key nodes of the 'threat network' (Supplementary Fig. 1A) were used as features.

The CMA and BLA masks were defined based on amygdala masks constructed by Shackman and colleagues[76]. These masks provide enhanced anatomical sensitivity and selectivity compared to the Juelich/SPM amygdala masks (see ref. 76). We added a coronal view of the amygdala masks (Supplementary Fig. 10) to enable readers to judge the alignment of the masks with the MNI152 template. Here, the CMA was defined by combining the central nucleus and medial nucleus. The BLA was defined by combining the lateral nucleus, basolateral and basomedial nuclei. The aHPC and pHPC masks were defined using the Harvard-Oxford subcortical probabilistic atlas (50% probability threshold, aHPC and pHPC were separated by Y = −21 mm in MNI space[77]). The dAI, vAI, and PI masks were defined based on clustering analysis of resting-state functional connectivity[78]. The vmPFC, dACC, and sgACC masks were created using Neurosynth[79] with "conditioning" as the keyword. An 8 mm sphere was created for each of the following identified peak coordinates: vmPFC (MNIxyz = −2, 46, −10), dACC (MNIxyz = 0, 14, 28), and sgACC (MNIxyz = 0, 26, −12). Voxels within the masks were concatenated as feature vectors for classification.

## The second analytic approach: decoding based on 'whole-brain' patterns
We tested recent conceptualizations that more distributed neural systems are engaged in threat processing[14,15,20–23]. We extracted voxel activations from the whole brain (restricted to the gray matter and excluded the 'threat circuit', Supplementary Fig. 1B) for the classification. To eliminate the impact of the 'threat circuit' on the decoding, voxels from the 'threat circuit', and their neighborhoods within a radius of 3 voxels, were excluded from the analysis.

## Classification procedure
We used logistic regression with L2-regularization (or ridge regression) from scikit-learn[80] as the classifier. The hyper-parameter of the L2-regularization term was selected from 20 equally distributed values between 0.01 and 100. The optimal hyperparameter was selected based only on training data. We assessed the classification performance on the discovery dataset using a 5-fold cross-validation procedure. Specifically, we evenly divided participants from the discovery dataset into 5 folds, trained the classifier (including hyper-parameter selection) using data from 4 folds of the participants, and tested the performance using data from the left-out participants. This procedure was repeated 5 times, with each fold of participants being the test set once. To avoid a potential bias of dataset split, we repeated the cross-validation procedure 10 times, using different splits in each repetition. Mean performance across all 10 repetitions was reported. We measured the decoding performance using forced-choice accuracy—a metric that is less sensitive to individual differences in overall activations and particularly suitable for fMRI analysis across sites[32–34]. In the forced-choice test, the classifier outputs of the two feature vectors of each left-out participant were compared, and the feature vector that led to a higher classifier output was chosen as CS + . The forced-choice accuracies were averaged across participants and reported.

## Generalization of the classifiers
We examined the generalization capability of the classifiers using validation datasets 1–2 which were from two studies with the same paradigm procedures and settings. We trained the classifier using all data from the discovery dataset and applied it to the two datasets that were never used during classifier training to evaluate the classification performance.

## Contribution of activation patterns to the classification

We examined the contribution of each voxel to the classification. We first extracted the model weights from the classifier trained based on the discovery dataset. We then transformed the model weights using the Haufe transformation to obtain the predictive weight of each voxel[38]. Differing from the model weight, the predictive weight is more reliable in representing the contribution of a voxel in the decoding. Furthermore, the predictive weights indicate the preference of voxel activations to the CS type—voxels with positive predictive patterns are more activated to CS+ (or CS+E/CS+U in extinction memory recall) presentation, while voxels with negative predictive patterns are more activated to CS- presentation. To increase the robustness of the estimated predictive weights, we repeated the procedure using the bootstrap resampling 10 times. The predictive weights were obtained by averaging values across the repetitions. We further conducted permutation tests for the whole-brain-based analysis to identify voxels that significantly contributed to the decoding. Specifically, we randomly shuffled the labels of the discovery dataset and re-estimated the predictive weight 1000 times, which generated the null distribution of predictive weight for each voxel. We then calculated Z scores and two-tailed $p$ values based on the mean and standard deviation of the null distribution. The significant voxels were identified based on false discovery rate (FDR) corrected $p$ values ($p < 0.05$, two-sided).

## Extended circuit

Based on the voxel contributions in the whole-brain-based analysis, we identified 14 brain regions (bilateral) that significantly contributed to the classification. With the P values from the permutation tests described above, we estimated the overall contribution of each voxel in the classification by averaging the mean ($-log(p)$) values across the 7 trial-blocks that showed robust classification performances[81]. We considered 'robust' classification as significant decoding accuracies in both discovery dataset and validation datasets 1–2 (Fig. 2A–C). All 4 trial-blocks of conditioning, the first 2 trial-blocks of extinction learning, and the first trial-block of extinction learning recall (both CS+E vs. CS- and CS+U vs. CS-) met this criterion. We did not include the second trial-block of extinction learning in the analysis, because the accuracy dropped a lot from the first to the second trial-block (both in the discovery dataset, and in the validation dataset 1), and the accuracies in the second trial-block for the two validation datasets were lower than 65% (64.9% and 60.3%). We then selected the top 10% voxels with the largest contributions. This choice was made because the mean percentage of voxels that significantly contributed to the classification ($p < 0.05$, FDR-corrected, two-sided) across the trial-blocks (Fig. 2D and Supplementary Fig. 6) was around 10% ($9.5\% \pm 2.2\%$). Also, the results were robust overall regarding the choices of percentage (Supplementary Fig. 7). Based on the distribution of these voxels, we identified 14 regions. All these regions except the OFC and CER were defined using the Harvard-Oxford probabilistic atlas (probability threshold: 50%). The OFC and CER were defined using the automated anatomical labeling (AAL) atlas. These 14 regions, together with the 10 traditional 'threat circuit' regions, comprised the updated threat detection and flexible responding circuit.

## Representational similarity analysis

To test the similarity of regional contributions to the decoding across the newly defined circuit, we performed a representational similarity analysis[42]. We estimated the inter-regional representational similarity based on the cross-participant correlation between the regional representation responses of every two regions, which resulted in a $24 \times 24$ symmetric matrix $R$. Conceptually, a high representational similarity value $R_{i,j}$ between two regions $i$ and $j$ suggests that they contributed to the classification similarly, while a low representational similarity value suggests that they contributed to the classification differently. More specifically, we first calculated the regional representation response for each participant. For a specific region $i$ and a participant $n$, given its classifier weight vector $w_i$ and the participant's activation vector to CS+ ($x_{n,i,cs+}$) and CS- ($x_{n,i,cs-}$), we calculated the regional representation response $Y_{n,i}$ using a dot product: $w_i*(x_{n,i,cs+} - x_{n,i,cs-})/v_i$, where $v_i$ is the voxel number of region $i$ to account for the effects of region size. A region with $Y_{n,i}$ larger than zero contributed more to code CS+ while a region with $Y_{n,i}$ smaller than zero contributed more to code CS-. We then calculated the Pearson's correlation between vector ($Y_{1,i}$, $Y_{2,i}$, ..., $Y_{N,i}$) and vector ($Y_{1,j}$, $Y_{2,j}$, ..., $Y_{N,j}$) across all $N$ participants as the representational similarity value $R_{i,j}$. After we obtained the representational similarity matrix $R$, we conducted the multidimensional scaling (MDS) with the number of components set to 2 to visualize the representational similarity between regions. In the MDS analysis, regions that contributed similarly to the classification were clustered together. For each trial-block, we identified two clear clusters, with one cluster composed of regions showing preferences for CS+ (regional representation response larger than zero) and another cluster composed of regions showing preferences for CS- (regional representation response smaller than zero). Across all trial-blocks, the regions were assigned to three communities that showed different patterns of preferences for CS types: consistently coded the CS+, consistently coded the CS-, or dynamically switched its preference to CS+/CS- depending on the trial-block.

## Validation on external datasets using different paradigms

Classifiers trained with brain activations of the extended circuit were tested on external datasets 3–9, which included threat conditioning paradigms, paradigms assessing 'subjective fear' to frightening images, as well as paradigms probing negative affects using intrinsically salient stimuli. These datasets were independently collected in other studies using paradigms different from the discovery dataset and the external datasets 1–2. We briefly describe the paradigms of these datasets here (summarized in Supplementary Table 4); more detailed descriptions can be found in the original publications.

External dataset 3 ($n = 299$) is from a visual conditioning task[44] using two male three-dimensional virtual humanoid characters as CS (one as CS+, the other as CS-). Each CS type was presented 16 times, with 8 of the CS+ presentations reinforced by an electric shock on the participants' wrist. The fMRI data were acquired using a 3 T GE MR scanner using an 8-channel head coil. The data were preprocessed using fMRIPrep 20.0.2, with the same settings as the discovery dataset. The GLM model included regressors for the reinforced CS+, unreinforced CS+, CS-, and shock. We used two regressors to model CS- trials, with one regressor containing the same number of CS- trials as unreinforced CS+ trials (8 trials), and the other regressor containing the remaining CS- trials (8 trials). This setting equalized the number of trials for unreinforced CS+ and CS- beta estimations. The beta maps for unreinforced CS+ and the corresponding CS- were used for validation.

External dataset 4 ($n = 94$) is combined from 3 different studies using visual conditioning paradigms[45]. The paradigms involved 2 CS+ (a face image and a house image) and 2 CS- (another face image and another house image). Each CS was presented 13 times, with 6 of the CS+ presentations reinforced by an electric shock on the participants' shin. The fMRI data were acquired using a 3 T Philips MR scanner using an 8-channel or 32-channel head coil. The data were preprocessed using fMRIPrep 20.0.2, with the same settings as the discovery dataset. We followed the same strategy as the original studies for this dataset in constructing the GLM, which resulted in 4 beta maps of interest (CS+ face, CS- face, CS+ house, CS- house). We separately validated the classifiers on two tasks: CS+ face vs. CS- face, and CS+ house vs. CS- house. The accuracies were averaged for the two classification tasks and reported.

External dataset 5 ($n = 48$) is from a visual conditioning task[43] that used two categories of objects as CS—one category as CS+, the other

as CS-. Pictures of 24 different tools and 24 different animals were presented, with 12 pictures of the CS+ reinforced by an electric shock on the participants' fingers on the left hand. The fMRI data were collected in a Siemens Skyra 3 T MR scanner using a 32-channel head coil. The data were preprocessed using fMRIPrep 20.0.2, with the same settings as the discovery dataset. The GLM model was similar to the external dataset 3, with beta maps for unreinforced CS+ and the corresponding CS- used for validation.

External dataset 6 ($n = 68$) is from an auditory conditioning task[46] that used two tones (800 or 170 Hz) as CS—one as CS + , the other as CS-. The CS+ was presented 16 times, with 8 reinforced presentations that paired with an electric shock to the right wrist, and 8 unreinforced presentations. The CS- were presented 8 times with no shock delivered. The fMRI data were collected in a 3 T Siemens Allegra MR scanner using a 32-channel head coil. Two beta maps (one for the unreinforced CS + , the other for the CS-) for each participant were obtained from the original study and used for validation.

External dataset 7 ($n = 65$) is from a study that investigated the subjective experience of fear[22]. Participants were presented with 80 different pictures and were instructed to report the fearful state they experienced for the stimuli from 1 (neutral/slightest fear) to 5 (strong fear). The fMRI data were collected in a 3 T GE MR scanner. The beta maps for each rating score (rating 1–5) were obtained from the original study and used for validation. We focused on classifying low vs. high fear, where features of low fear were obtained by averaging beta maps of ratings 1 and 2, and features of high fear were obtained by averaging beta maps of ratings 4 and 5.

External dataset 8 ($n = 182$) is from a study that examined picture-induced negative affect[34]. Participants were presented with 15 neutral and 15 negative pictures and were instructed to report their emotional state using a 5-point Likert scale (1 for neutral, 5 for strong negative). The fMRI data were collected in a 3 T Siemens Trio MR scanner using a 12-channel head coil. The beta maps for each rating score were obtained from the original study. We focused on classifying low vs. high negative states, where features of low negative states were obtained by averaging beta maps of ratings 1 and 2, and features of high negative states were obtained by averaging beta maps of ratings 4 and 5.

External dataset 9 ($n = 59$) is from a study that examined the neural representations for physical pain and social pain[19]. The participants underwent somatic pain and social rejection tasks. In the somatic pain task, participants received heat (painful) or warm (non-painful) thermal stimuli that were delivered to their left volar forearm. In the social pain task, participants were presented with photographs of their ex-partner or a close friend, and they were instructed to think about the break-up experience or positive experience, respectively. The beta maps for each experimental condition were obtained from the original study. We focused on classifying warm vs. heat for the somatic pain task and friend vs. ex-partner for the social rejection task.

To compare the decoding accuracies across different paradigm categories, we divided the external datasets 3–9 into 3 groups: conditioned threat (datasets 3–6), 'subjective fear' (dataset 7), and intrinsically salient stimuli (datasets 8–9). For each group, we calculated the proportion of participants whose beta maps were correctly decoded across datasets. And then used the Chi-square test to compare the proportion between every two dataset groups. This analysis was separately done for the model of each trial-block, with the obtained p-values adjusted based on false discovery rate (FDR) correction.

### Statistical analysis

Permutation tests were used to determine the statistical significance of the classification performance. Specifically, we randomly permuted the labels of the activation maps (CS+ or CS-) 1000 times. We conducted the cross-validation procedure as described above using the permuted labels to obtain the null distribution of the classification accuracy. The $p$ value was calculated by counting the percentage of accuracies within the null distribution that was higher than the accuracy obtained with the real data (i.e., one-sided, since we care about accuracies higher than chance level). For external validation datasets 1–2, we trained the classifier using the label-permuted discovery dataset and applied it to the external datasets to obtain the null distribution. For external validation datasets 3–9, we used two-sided binomial tests to evaluate the significance of the forced-choice accuracies[32–34].

### Reporting summary

Further information on research design is available in the Nature Portfolio Reporting Summary linked to this article.

## Data availability

The validation dataset 2[37] is available at NIMH Data Archive through collection ID 2393. The validation dataset 4[45] is available in OpenNeuro database with the following accession numbers: ds003550, ds003553, and ds003554. The validation dataset 5[43] is available at https://doi.org/10.17605/OSF.IO/QEG83. The validation dataset 6[46] is available at https://doi.org/10.17605/OSF.IO/68YWZ. The validation dataset 7[22] is available at https://doi.org/10.6084/m9.figshare.13271102.v2. The validation dataset 8[34] is available at https://identifiers.org/neurovault.collection:503. The validation dataset 9[19] is available at https://github.com/cocoanlab/interpret_ml_neuroimaging. The discovery dataset (including data from published[25,26,69–73] and unpublished studies), and the validation datasets 1[36] and 3[44], are available upon request due to the need to establish data sharing agreements. There are no restrictions to who the data can be made available to, and there are no restrictions for data use for research. Requests for discovery dataset should be directed to M.R.M (mohammed.r.milad@uth.tmc.edu). Requests for the validation dataset 1 should be directed to S.W.L. (slazar@mgh.harvard.edu). Requests for validation dataset 3 should be directed to F.A. (fredrik.ahs@miun.se). The regional masks, predictive patterns, and trained classifiers are available at https://github.com/zhenfu-wen01/threat-mvpa and in a Zenodo repository at https://doi.org/10.5281/zenodo.10452949[82]. Source data are provided with this paper.

## Code availability

The codes and data to generate the main figures and results are available at https://github.com/zhenfu-wen01/threat-mvpa and in a Zenodo repository at https://doi.org/10.5281/zenodo.10452949[82].

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

## Acknowledgements

This work was supported by the National Institute of Mental Health grants R01MH123736, R01MH125198, R33MH111907, R01MH097880, and R01MH097964 to M.R.M. R01MH109638 to E.F.P.-S. and 1R01AT006344 to S.W.L.

## Author contributions

Z.W., E.A.P., J.E.L and M.R.M. conceived and designed the study. E.F.P.-S., S.W.L., J.R. and F.A. provided data for analyses. Z.W. analyzed the data under the supervision of M.R.M. Z.W. and M.R.M. interpreted the data with input from all authors. Z.W. and M.R.M. drafted the manuscript. All authors revised the manuscript and approved its final version for submission.

## Competing interests

Praxis Precision Medicines, Inc. provided partial salary support to E.F.P.-S. The remaining authors declare no competing interests.
