## [Peer Review File · Nature Communications]

Distributed neural representations of conditioned threat in the human brainReviewer #1 (Remarks to the Author):

The authors investigated neural representation of threat and safety across different tasks and experimental designs, with particular focus on integration across brain regions beyond the traditional threat circuit. They convincingly show that regions beyond the threat circuit significantly contribute to processing of threat and safety, and that different regions tend to be associated with threat, safety and adaptive switching between the two. Importantly, the authors present results from a discovery as well as two validation data sets, which is a great practice towards generalizable science. The methodology used is robust, though I have some concerns and questions which I present below.

1.

"A first-order autoregressive model was used to account for the temporal structure of the noise"

a) What time points were the classifiers trained on? b) How did the authors ensure that the outcome from the previous trial was not included in the decoded signal? c) The AR(1) model is known to perform poorly at removing autocorrelation (Mumford et al. 2012; <https://doi.org/10.1016/j.neuroimage.2011.08.076>), this is particularly important at dissociating the actual outcomes (shocks, pain etc) from threat. The authors should either demonstrate lack of such signals or take thorough steps to remove the outcome signal.

2.

"Considering the dynamic nature of the threat/extinction learning processes, we divided each experimental phase into 4 trial-blocks (TB, from TB1 to TB4) and built a decoding model for each trial-block separately."

The number of trials that went to each trial block should be clarified in the introduction/results - is it always 4? Are there any differences between studies in the discovery dataset and/or between discovery and validation datasets?

3.

CS+E and CS+U in extinction seem to both drop off, however, I would expect that CS+U would remain higher since it remained threatening during extinction. How do the authors interpret this, especially since in the discovery set and in validation set 1 CS+E seems to be higher than CS+U (in threat circuit analysis).

4.

It is a bit unclear how the data sets were selected - are these all available published data sets using the Milad et al. paradigm? If not, what was the selection process?

5.

In Figure 2D, why does extinction only have 1-4 CS? The number of trials is the same as acquisition (p16).

6.

It only becomes clear in discussion that the discovery dataset uses exclusively Milad et al. task - add to introduction/results.

7.

"The optimal hyperparameter was selected based only on training data"

Please share the results of the L2-regularization tuning, including the final value used in all analyses.

8.

(introduction)

"Past research within this field has primarily focused on the roles of a few subcortical and cortical

structures in how the association between the conditioned and the unconditioned stimuli is formed, and how the defensive responses are generated and subsequently extinguished.” Perhaps since authors are referring to the past research into Pavlovian threat reactions, they should also refer the reader to the Fullana (2016) article <https://www.nature.com/articles/mp201588>.

9.

(introduction)

“For example, the amygdala is a key node in the acquisition and extinction of conditioned threat; and the hippocampus is critical for contextual information processing”

Please cite.

10.

Please share analysis code together with reproducibility instructions, as well as the resulting safety/threat maps.

11.

Reference 7 contains full first name

Reviewer #2 (Remarks to the Author):

#####

Distributed neural representations of conditioned threat in the human brain

Wen...& Milad

#####

Despite its basic and translational importance, the neurobiology of fear remains incompletely understood, impeding the development of more effective interventions for maladaptive fear. Most human imaging studies employ small samples and conventional mass univariate analyses, which have important inferential limitations. Thus, the focus of this paper is timely and likely to be of interest to a broad spectrum of researchers and clinical scientists.

Key Strengths

- extraordinarily large sample for this domain of research, with the potential for enhanced power, precision, and reproducibility
- fairly comprehensive tests of sensitivity, specificity, and generalizability via multiple independent hold-out datasets (ie. not used for classifier training)
- the innovative MVPA classifier approach has the potential to address questions of specificity that cannot be addressed using traditional mass univariate approaches, innovation is strengthened by the thoughtful bin-wise approach, which provides important clues about temporal dynamics
- the first author (Dr. Wen) is to be congratulated on what amounts to a truly enormous amount of work (e.g. data wrangling, processing, and analyses for 9 datasets)

Key Weaknesses

- the Introduction is unclear and occasionally (mildly) misleading
- Some of the claims in the Discussion go beyond the data and analyses

In the section that follows, I provide a few suggestions for further strengthening the manuscript.

MAJOR / GENERAL

1. INTRODUCTION.

a. The study (while useful, significant, and innovative) is poorly motivated in the introduction. e.g.

"Although largely supported by cross-species research, this simplified model based on 62 Pavlovian reactions does not fully account for the range of complex cognitive and 63 memory representations that are formed and stored in distributed neural circuits and 64 used to guide deliberative defensive actions. While the 'threat circuit' is highly 65 necessary for the detection and response to threat, there is a need for studying how this 66 circuit interacts with other cognitive and sensory processing neural networks^{10,11}. 67 Currently, we do not have established distributed neural decoders that are integrative, 68 sensitive, and reproducible for the detection and response to threat and safety. To move 69 beyond simplified circuits of threat detection and reaction, we examined the complex 70 neural circuits predictive of Pavlovian threat learning throughout the human brain to 71 reveal a more integrated and updated conceptualization that highlights the dynamic 72 interplay between multiple neural systems to detect the threat, and to form implicit and 73 explicit memory representations for prospective defensive actions."

Nothing about "circuits," "networks," "how...[the threat-anticipation] circuit interacts with other...networks," or "the dynamic interplay between multiple neural systems" motivates the development of a brain signature based on sluggish BOLD responses to Pavlovian cues. There are excellent reasons for using MVPA, but these are not them.

[Note: This concern remains true even when considering the block-wise RSA analyses]

b. A key innovation of the approach is the focus on extinction (day 1) and extinction recall (day 2), but this appears "out of the blue" several hundred words into the introduction. I would encourage the authors to mention/foreshadow this earlier. This is particularly important in light of evidence suggesting that extinction is likely more relevant to pathological fear and anxiety.

c. Based on the introduction, it's not clear what the 'second analytic approach' entails. e.g.

"we investigated the contribution of distributed neural 84 systems in the formation of memory representations at higher cortical levels, which may 85 be related to encoding deliberative defensive actions."

As a consequence, it's not clear what is meant by the next section:

"From the combined first and second analytic approaches, we constructed an 92 extended circuit that includes sensory, working memory, and cognitive neural nodes. 93 We trained new classifiers based on neural activations from this extended circuit 94 obtained from our threat conditioning datasets,"

[are the authors referring to the 'virtual lesion' approach described in 'Decoding based on w-b patterns?' ... cf. Fig. S1b]

d. Minor/Moderate - A number of minor claims made "in passing" are potentially misleading to readers who are either unfamiliar with this literature, or who are only familiar with the animal or imaging arms of it. See my minor/specific comments, below.

e. Some of the terminology/nomenclature drifts across the paper. Please revise for consistency.

2. METHOD.

a. Participants - Authors should add a table to the main report or Supplement with basic demographic and scanning information for the studies that contributed to the 1,465. Authors should clearly indicate whether each study was used for training/discovery vs. hold-out testing. Something similar should be done, at least in the Suppl, for the "external datasets." Here, the authors should briefly indicate the nature of the paradigm employed in each external dataset.

b. There is drift in the terminology/nomenclature. Study vs Setting. The Method section does not use the First/Second Analytic Approach terminology. Please revise for consistency across the entire paper. Moving some of the imaging parameter details to tables (see above) might make this easier for readers to digest.

c. There are longstanding concerns that the Juelich/SPM amygdala masks are poorly aligned with the probabilistic MNI152 templates (e.g. Tillman et al. Hum Brain Mapp 2018 Figure S5). It is not clear based on Fig S1a whether this is a concern here. Authors should provide a coronal slice in the supplement to enable readers to judge for themselves. In the event that inspection reveals inadequate alignment, authors should appropriately modify the approach.

d. Reducing vmPFC, sgACC, and MCC/dACC to 8-mm spheres (cf. Fig S1a) is an unexpected choice; this means that the vast majority of voxels in those 3 cortical regions cannot contribute to the classifier. It also means that some of the "other" regions (e.g. hipp, aIns) will actually contribute more measurements to the classifier training than these ostensibly "larger" cortical regions. Furthermore, it also means that many, many vmPFC and MCC/dACC voxels that fall outside of the a priori masks will contribute to the "whole-brain" classifier, with potentially significant implications for interpretation (i.e. the Discussion is framed around broad regions, such as "dACC"). I do not think this curious choice is inherently problematic, but I would encourage the authors to appropriately motivate this choice in the Method section for curious readers.

3. ANALYTIC STRATEGY.

4. RESULTS.

a. The Haufe-transformed weight shown in fig 1d are presented in such a superficial way in the main text that they add little to the paper. These are really quite interesting! Authors are strongly encouraged to adequately describe them in the main text - walk the reader thru which regions prefer threat, which prefer safety, and which shown bin-dependent "dynamics" (but see my comments about the Discussion, below).

b. The virtual lesion results are interesting, but the verbal framing seems to muddle 2 ideas: (a) maximizing predictive performance using more voxels versus (b) the degree to which the 'threat-anticipation circuit' (ie the usual suspects) is critical for decoding performance. (This is also something of an issue in the Discussion; see below).

"132 Although significant, the decoding accuracies based on the 'threat circuit' activation
133 patterns were often below 75%. Given the conceptual framing stated above, we
134 reasoned that considering the neural representations from multiple sensory, attention,
135 and cognitive circuits should improve the discrimination accuracy of the classifier.
136 Indeed, using neural representations from the entire brain (excluding the 'threat circuit')
137 led to substantially improved decoding performances (Figure 2A)."

I would encourage the authors to refine the framing to sidestep muddling. They *might* also wish to add a supplementary analysis that uses *all* GM voxels (i.e. the logical-AND conjunction of the voxels used in figure 1 and figure 2), as this would answer an obvious question many readers will have.

c. Figure 3a seems deeply misleading. See my prior comment about using 8-mm spheres to sample vmPFC, dACC/MCC, etc. Please revise to make it clear that some regions are reduced to small spheres. Also - it is difficult to see the division between CM vs. BLA ... please use a darker/bolder border as with the other regions (or the broken line used in vmPFC-vs-SGACC)

d. "Based on the regions that exhibited high contribution to the decoding models, and the 160 traditional 'threat circuit', we constructed an 'extended threat detection and responding 161 circuit' (Figure 3A), which, based on our data, better defines the brain regions involved 162 in threat processing than the 'threat circuit' alone."

Seems odd to jump to RSA, before describing performance for the "extended" classifier/decoder.

e. "The first community (red color in Figure 3A) consistently coded for threat (CS+) across 167 conditioning, extinction learning, and recall. The second community (blue color in 168 Figure 3A) consistently coded the safety of the cue (CS-). The third community (yellow 169 color in Figure 3A) showed dynamic coding; shifting their signal between threat and 170 safety (CS+ or CS-) depending on the experimental phase."

Please summarize the regions in each community in the main text - thanks!

5. DISCUSSION / IMPLICATIONS.

a. Some key interpretive claims need to be unpacked. For instance, one of the major take-home points is that the present results reinforce theoretical work underscoring the importance of regions beyond 'the usual suspects' for processing threat, but this is simply claimed and tagged with a list of numeric references. It needs to be briefly unpacked, and linked to the theoretical literature. See my minor/specific comments below for details.

b. A major take-home point is that the decoders work quite nicely even when you "virtually lesion" all of the usual suspects, but this is barely mentioned at all in the Discussion.

c. Authors need to be more precise in their language. Reliable, critical, identify/detect, and respond have specific meanings that do not correspond to how they're used in the paper. See my minor/specific comments below for details.

d. "In contrast, multiple brain regions, e.g., the dorsal 273 anterior insular cortex and the dorsal anterior cingulate cortex (dACC) consistently 274 contributed to the classification of the CS+, which support the role of these regions in 275 the encoding of the associative aspect related to threatening cues and/or to the 276 generation of conditioned threat responding."

This interpretative claim goes well beyond that licensed by the data and approach. There is nothing in the results shown in fig 1d and 3a that rules out the possibility that these specific regions are equally (or more) responsive to intrinsically/unlearned aversive stimuli [i.e., the tests are whole-decoder, not region specific]. And it is certainly the case that these regions respond (in a mass univariate sense) to a broad range of stimuli (cf. meta-analyses by De La Vega...& Wager/Yarkoni; Chang...& Yarkoni; Shackman...& Davidson; and work by Uddin/Pessoa). Please fix the claims to align with the results, or vice versa. The authors *might* consider adding some supplementary analyses (e.g. x-validated x-classification) to determine whether these specific regions are preferentially/differentially associated with pavlovian threats, but this is not essential.

e. "On the other hand, several neural nodes showed transient preferences in the encoding 284 of the CS+ and CS- across experimental phases...like the MFG, SFG, vmPFC, and

286 amygdala, amongst others."

While the CeMe, BLA, and vmPFC results shown in fig 1d support this claim, it wasn't clear to me that any of the results shown in any of the figures support the claim about MFG/SFG (dIPFC). Please revise the Discussion to match the Results (or vice versa).

Also, the results differ across the CeMe vs BLA roi's (cf. fig 1). This distinction should be clearly communicated here in the Discussion, not lumped.

f. "We refer to those neural nodes as 'flexible' coders. Why do 287 they change their encoding of the CS type?"

I may have missed it, but the authors do not seem to formally test whether "they change" and it is not clear which of the color patches shown in fig 1d are actually significant. In some cases (e.g. CeMe, vmPFC), visual inspection suggests something more like habituation than "changing" or "flipping." Consider the vmPFC sphere. Muted red at the onset of acquisition (significant?) ...then "nothing" ... then dark-blue at the onset of extinction ... then muted blue for the remainder. As it stands, I'm not sure whether this is "even" interpreting the null. Please revise the Discussion to match the Results (or vice versa).

g. The authors do not really address the larger significance of the decoders. Now that they exist and have been validated, how might they be applied. Note: I strongly encourage the authors to share their models with the scientific community via Neurovault, OSF, Github, or another public repository. This maximizes their value and increases the likelihood that they will be further developed and refined.

h. Authors need to very briefly acknowledge limitations of the study (e.g. the gap between predictive multivoxel models and causal insights licensed by mechanistic studies); these could even be framed as key challenges for future research.

6. FIGURES.

- figure 1 ... it is not intuitively clear that the columns in panel d indicate bins. this should be clearly labeled.

For Brain Imaging Papers

- Is the length of each trial and interval between trials specified?

useful to provide pavlovian cue duration(s)

- Is the field strength (in Tesla), pulse sequence type (gradient/spin echo, EPI/spiral), field-of-view, matrix size, slice thickness, and TE/TR/ flip angle clearly stated?

useful to provide at least some of these details in tabular form for the studies used for training and testing ... can skip field strength if all 3T; can skip sequence type and FOV; can report voxel size; can skip TE and flip angle; but should report TR

- If there was data normalization/standardization to a specific space template, is the type of transformation (linear vs. nonlinear) used and image types being transformed clearly described?

not clear; it's not sufficient to just say fmripred defaults. both the normalization and T2*-T1 co-reg should be described; the specific MNI atlas/template should be noted.

MINOR / SPECIFIC

- the References are a bit of a mess; please carefully copy edit for consistency

- "Past research

51 within this field has primarily focused on the roles of a few subcortical and cortical
52 structures in how the association between the conditioned and the unconditioned stimuli
53 is formed, and how the defensive responses are generated and subsequently
54 extinguished."

this is misleading, at least in terms of the human imaging lit; 'intensively' focused or similar would be better aligned w/ the state of the science

- "This approach has led to the notion of the so-called 'threat circuit' that
55 mainly includes subregions of the amygdala, periaqueductal gray, hippocampus, medial
56 prefrontal cortex, and insular cortex^{3,7-9}."

would consider adding Shackman & Fox AJP 2021 &/or Fullana's fear signature meta-analysis

also - given that the authors focus their analyses on the MCC/dACC, it might make sense to replace 'medial pfc' with 'cingulate' or similar

- "This approach has led to the notion of the so-called 'threat circuit' that
55 mainly includes subregions of the amygdala, periaqueductal gray, hippocampus, medial
56 prefrontal cortex, and insular cortex^{3,7-9}. This concept suggests the functional specificity
57 of these regions during the acquisition and extinction of conditioned threat^{3,4}. For
58 example, the amygdala is a key node in the acquisition and extinction of conditioned
59 threat; and the hippocampus is critical for contextual information processing."

hard to follow the narrative thread, as written; the first sentence is clear and true to the mechanistic and imaging literatures...but it is not 100% clear what's meant by "suggests" ... do the authors mean that the existence of a 'threat-anticipation circuit' concept tempts scientists to over-interpret, and conceptualize these regions as *specific* to fear/anx (a reverse inference type fallacy? my confusion is amplified by the "for example" sentence, which seems focused on mechanistic evidence of causal necessity. please revise this section for clarity

- "Although largely supported by cross-species research,"

this seems potentially misleading/disingenuous. as the authors well know, the Fullana meta-analysis failed to reveal consistent/significant amygdala activation in human FCing studies; while the authors did demonstrate early amygdala activation in their recent PNAS report, this evidence is not specific to FCing, given the nature of the contingencies at the outset of the acquisition scanning.

now, i recognize that this is the brief introduction to an empirical report -- not a comprehensive review -- so there's no need to get into all of the weeds, but this lack of consistency across species/methods needs to be acknowledged in passing. for instance, the authors could modify the clause to indicate

"Although largely supported by cross-species research Ref1 (but see Ref2),"

- "guide deliberative defensive actions"

"we investigated the contribution of distributed neural
84 systems in the formation of memory representations at higher cortical levels, which may
85 be related to encoding deliberative defensive actions."

do you really mean to say 'deliberative' ? this seems unduly speculative for the introduction.

- "While the 'threat circuit' is highly
65 necessary for the detection and response to threat"

this seems misleading, insofar as not every node in that network appears to be mechanistically critical

- "To understand the
79 specific value of the 'threat circuit' to the detection of threat, we first employed MVPA to
80 examine neural patterns within this circuit in distinguishing stimuli that have been
81 conditioned to signal threat from those that do not."

in the absence of behavioral evidence of "detection" this seems misleading; please revise to better align with the strengths and limitations of the data and approach

- "In the second analytic approach,
82 based on recent conceptualizations that more distributed neural systems are engaged
83 in threat processing10,11,13-17,"

"We tested recent conceptualizations that
453 more distributed neural systems are engaged in threat processing10,13,14,25,67."

"Our results
336 strongly support recent conceptualizations calling for the integration of broad networks
337 required for the implicit and explicit processing of threat."

authors may consider citing this lovely new paper - up to you

Review Neurosci Biobehav Rev

. 2023 May 18;105237. doi: 10.1016/j.neubiorev.2023.105237. Online ahead of print.

The Nature and Neurobiology of Fear and Anxiety: State of the Science and Opportunities for Accelerating Discovery

Shannon E Grogans 1, Eliza Bliss-Moreau 2, Kristin A Buss 3, Lee Anna Clark 4, Andrew S Fox 2, Dacher Keltner 5, Alan S Cowen 6, Jeansok J Kim 7, Philip A Kragel 8, Colin MacLeod 9, Dean Mobbs 10, Kristin Naragon-Gainey 11, Miquel A Fullana 12, Alexander J Shackman 13

Affiliations expand

PMID: 37209932 DOI: 10.1016/j.neubiorev.2023.105237

also - why are the citations slightly different across the 2 quoted sentences?

- "1465"  1,465; same for "1000"

- "Participants. We analyzed data from a total of 1465 participants across multiple
345 studies. Participants from the discovery dataset (N = 425) were combined from multiple
346 studies18,20,55,56 and the data from the external datasets 1 (N = 98) and 2 (N = 127) were
347 from two other studies29,30."

authors should provide a table with basic demographic and scanning information for these studies

e.g. citation, N, % F, mean age, summary of diagnostic status as relevant, basic fMRI details ,and whether each study was used for training/discovery or hold-out testing

- minor - it would be useful to know the duration (or range of durations) for the pavlovian cues

- very minor - rationale for using $\gamma = -21$ for splitting anterior from posterior hipp should be provided

- "We then transformed the 464 model weights as suggested in a previous study³¹ to obtain the predictive weight of 465 each voxel."

authors might explicitly term this "Haufe-transformed"

as in this recent report from Avram Holmes & BTT Yeo

- "Validation on external datasets using different paradigms. We tested the sensitivity 531 and specificity of the newly established circuit in identifying conditioned threats on 532 external datasets 3-9."

hard to follow a written; by which i mean that Luke's IAPS dataset - for instance - does not include conditioned threats; please revise for clarity

- "To compare the decoding accuracies across different paradigm categories, we divided 602 the external datasets 3-9 into 3 groups: conditioned threat (datasets 3-6), 'subjective 603 fear' (dataset 7), and emotionally salient stimuli (datasets 8-9)."

pavlovian threat cues are emotionally salient; better to say something like 'naturalistic'

- "The 117 'threat circuit'-based decoding models were successfully generalized to the two external 118 datasets (Figures 1B, 1C)."

authors should summarize the results in the main text, as they did in the earlier section

- "stayed 145 even"

use formal English please; please polish

- "The improved classification performance of the 'whole-brain' pattern over the 'threat circuit' stayed 145 even when the number of voxels was matched (Figure S2)."

"improved"  numerically superior

[note - i may have missed it, but i do not think the authors formally tested the difference]

- "We used representational similarity 163 analysis³³ to test the contribution of each node within this updated circuit to the 164 decoding of threat and safety (Supplemental Figure 2)."

typo? wrong fig cited?

- "Lastly, we investigated how reliable and specific"

"More importantly...in a reliable and reproducible 262 manner—classifiers based on activation patterns from these nodes can be generalized 263 to different conditioning paradigms."

"provides a cohesive integrative extended threat

334 circuit that encodes for threat and safety, consistently, specifically, and reliably"

"reliable" has a specific meaning in the psychometrics lit; please avoid using it here; these analyses gauge convergent/divergent validity ("specificity") not "reliability"

- "we investigated how reliable and specific the extended circuits are when 180 classifying threat-conditioned stimuli across different paradigms"

isn't it singular "circuit"? or do you mean circuit(s)/model(s) because they are trained separately on each bin? please revise for clarity and consistency. perhaps makes sense to use a term such as "extended threat circuit decoders" or "binwise threat circuit decoders"

- "Some of these datasets 184 included variant Pavlovian conditioning paradigms, and others included negative affect 185 paradigms."

drift in nomenclature; sometimes described as 3 sets of dataset; here described as 2, ignoring the "subjective fear" distinction

- "We first tested the optimized models"

extended models

- "visual cues but assessed 'subjective fear'17,"

revise to make it clear that these are not mere 'visual cues' but are photos of ostensibly frightening animals/scenes

- "emotionally salient stimuli"

pavlovian threat cues are emo'ly salient ... this should revise to make it clear that there are intrinsically aversive stimuli

- "not related to associative learning and memory."

this is potentially confusing as written; of course, Ss learn about and form memories of these cues...just say something about unlearned/intrinsically aversive

- figure s2 - are the x-axis labels artifactually compressed?

- figure s3 - text labels (mni coordinates) are too small to read; please revise

- "The accuracies of the classifiers were reduced when decoding neural representations 206 about picture-evoked negative affect28"

"reduced"  "numerically lower (but still significantly above chance)"

- not clear why the authors do not show the grey "zone" corresponding to chance performance in figure 3b-3i; also the colors used for the box-and-whiskers is not consistent with the earlier figures where, for example, green = extinction recall ... please revise for consistency

- "most 209 of the models showed reduced accuracies for physical pain"

most of the decoders showed numerically reduced accuracies for physical pain

- "Decoding accuracies for conditioned threat

214 (datasets 3-6) were comparable to those for 'subjective fear' (dataset 7) (Chi-square
215 test, TB1-TB4: all $\chi^2 < 5.2$, $p > 0.08$), and significantly higher than those for emotionally
216 salient stimuli (datasets 8-9) at trial-blocks 2-4 (TB1-TB4: $\chi^2 = 1.1$, $p = 0.31$; $\chi^2 = 47.5$, p
217 < 0.001 ; $\chi^2 = 30.6$, $p < 0.001$; $\chi^2 = 55.6$, $p < 0.001$). Overall, these results support the
218 specificity of the decoding models to the neural correlates of stimuli associated with the
219 conditioned threat."

no they do not - Becker's fear photos work as well as Milad's conditioned visual threat cues; please
revise to align with the actual results

- "The trained classifiers are sensitive and reproducible regardless of experimental
11
228 paradigms or fMRI scanners."

you could go a step further and mention across countries ... data are collected in different continents!

- "Our results continue to highlight the critical role of the
229 'threat circuit' in the detection and response to threat."

"In addition to the 'threat circuit', our data highlight the value of other critical nodes that
253 are also important in detecting and processing threat"

"More importantly, we demonstrated that these neural nodes contribute to
261 the detection and processing of threat and safety cues in a reliable and reproducible"

these are observational/recording, not mechanistic/perturbation data; please don't use the term
'critical' ... likewise, these results speak to "classification" not detection or orchestrating defensive
responses

- "Importantly, our results advance
230 current conceptions in the field by showing that sensory and cognitive neural nodes are
231 also essential in enhancing the classification accuracies related to threat and safety
232 processing."

how does numerically increased classification accuracy advance or refine theory? authors need to
unpack the claim

- "In addition to the 'threat circuit', our data highlight the value of other critical nodes that
253 are also important in detecting and processing threat. When we encounter threat, we
254 need to process all sensory modalities currently present, pay attention to our
255 surroundings, retrieve prior memories related to the encounter of this or similar threats,
256 and decide how to respond. Indeed, our results do show a critical engagement of
257 multiple sensory and cognitive systems during the encounter of threat and safety cues,
258 which is likely needed for the processing of these complex representations. These
259 regions spread across the primary sensory and motor cortex, frontal-parietal cortex, and
260 cerebellum."

it is perhaps worth noting that the present results provide compelling new empirical evidence for
theories that make exactly this point -- cf. "Question 3. Afterword" in Grogans et al...

Review Neurosci Biobehav Rev

. 2023 May 18;105237. doi: 10.1016/j.neubiorev.2023.105237. Online ahead of print.
The Nature and Neurobiology of Fear and Anxiety: State of the Science and Opportunities for Accelerating Discovery
Shannon E Grogans 1, Eliza Bliss-Moreau 2, Kristin A Buss 3, Lee Anna Clark 4, Andrew S Fox 2, Dacher Keltner 5, Alan S Cowen 6, Jeansok J Kim 7, Philip A Kragel 8, Colin MacLeod 9, Dean Mobbs 10, Kristin Naragon-Gainey 11, Miquel A Fullana 12, Alexander J Shackman 13
Affiliations expand
PMID: 37209932 DOI: 10.1016/j.neubiorev.2023.105237

- "and the dorsal anterior cingulate cortex (dACC)"

why is this acronym redefined in the discussion; please fix

- "Our analyses of voxel contributions to the decoding of CS+ and CS- across 269 experimental phases revealed interesting predictive patterns (summarized in Figure 3a)."

more appropriate to cite figs 1d *and* 3a

- "These brain regions continued to signal, or 277 encode, the CS+ even during extinction learning and during extinction memory retrieval, 278 consistent with maintaining some representation of the initial conditioned threat memory 279 regardless of extinction. These data provide neurobiological support from the human 280 brain to the now well-accepted concept that extinction does not erase the original 281 conditioned threat associations, first proposed by Pavlov in 192753."

This section needs some nuance. In fact, fig 1a reminds us that decoder accuracy drops to chance levels for most of extinction (bins 2-4) and recall with extinction (bin 2). Please revise to better convey this nuance.

[Aside - As noted above, the extinction and recall data are real strengths of the paper, and I think these results are cool as-is.]

- "Despite all of these variations, the models trained on one paradigm were still 312 highly sensitive and specific in distinguishing threatening from non-threatening 313 conditioned cues across paradigms, suggesting that the classifiers learned some 314 general patterns of conditioned threat."

please cite fig 3b-3d

- "were applied to studies that used auditory fear conditioning, supporting the idea that"

only 1 study, not studies

please cite fig 3e

- "And as expected, the accuracies of our models tested in the 322 negative affect and pain processing studies were substantially lower compared to those 323 obtained from the threat conditioning studies."

significantly lower

cite fig 3g-3i

consider mentioning the chi-square tests (which support the claim of 'significantly lower')

i.e. "Decoding accuracies for conditioned threat

214 (datasets 3-6) were comparable to those for 'subjective fear' (dataset 7) (Chi-square
215 test, TB1-TB4: all $\chi^2 < 5.2$, $p > 0.08$), and significantly higher than those for emotionally
216 salient stimuli (datasets 8-9) at trial-blocks 2-4 (TB1-TB4: $\chi^2 = 1.1$, $p = 0.31$; $\chi^2 = 47.5$, $217 <$
0.001; $\chi^2 = 30.6$, $p < 0.001$; $\chi^2 = 55.6$, $p < 0.001$). Overall, these results support the
218 specificity of the decoding models to the neural correlates of stimuli associated with the
219 conditioned threat."

- "In summary, our results contribute to the expansion of the 'threat circuit' by including
331 multiple other sensory and cognitive neural nodes."

do they? authors need to carefully explain the value of mvpa classifiers for inference in the
introduction, and they need to be more precise here, in the discussion, about what these results
mean. if the goal is accurate decoding of pavlovian conditioned threat, you need to include regions
well beyond 'the usual suspects' ... but those 'extended' regions are also less likely to be specific to
'threat' ... as it stands, the degree to which 'traditional' vs. 'extended' regions show specificity across
tasks is not addressed by the present results, so more care needs to be taken in how the results are
framed

- "provides a cohesive integrative extended threat
334 circuit that encodes for threat and safety, consistently, specifically, and reliably"

this claim goes well beyond the data; there is nothing about MVPA (regularized multiple logistic
regression) that guarantees "cohesive/integrative" ... it's blind statistical prediction of cue-type ...
please fix

- Authors are strongly encouraged to add an Author Contributions section to the paper or supplement

Reviewer #3 (Remarks to the Author):

Wen and colleagues investigated distributed brain signals linked to conditioned threat in the human
brain. This is an excellent study that provides a clear contribution to the field. It addresses a timely
and important question concerning the existence of a spatially distributed, "extended threat" circuit.
The paper is well written and the methods appear sound. I have only a few points of clarification that I
outline next:

To define the extended circuit, the authors adopted a series of criteria that, while reasonable, were ad
hoc. The authors state that 7 blocks with "robust" classification were used. How was "robust" defined?
Then, 10% of voxels with the largest contributions were used. While a more thorough analysis of how
these choices impact the results is perhaps beyond the scope of the paper, it would be important to
provide some sense of the overall stability of the results in terms of these choices. For example, if the
extended circuit looks quite different with 20% of the voxels, what would that mean?

In this context, I feel torn about the inclusion of the OFC. The first time I noticed that it had been
treated differently was in the Methods, but maybe I missed it before. I think the text of the Results
and then the Discussion should discuss this more explicitly so that it's not buried in the Methods (but I
might have missed it).

I'm having some trouble connecting the extended circuit so nicely summarize in Fig 3A to the results
in Fig 2D. My suggestion to help the reader better appreciate the relationship would be to have more
transparent brains in 2D with the outlines of the regions shown in 3A on them. In any case, I think it's

important to help the reader get a better appreciation of the contributions of the extended circuit in the actual brain maps.

Minor:

ROIs: hopefully the authors will share their ROI masks so that others can also employ them, but an additional sentence or two should indicate more clearly how the CMA/BLA masks were defined.

Dear Editor,

We herein submit our revised manuscript, titled “Distributed neural representations of conditioned threat in the human brain” by Wen et al., to be considered for publication at Nature Communications. We are grateful to the reviewers and editor for their careful read of the paper and their insightful and constructive comments. We carefully considered and addressed each comment, and made extensive revisions in the manuscript accordingly. Our point-by-point response is stated below. The changes we made in the manuscript were marked red.

Reviewer #1:

General comment: *The authors investigated neural representation of threat and safety across different tasks and experimental designs, with particular focus on integration across brain regions beyond the traditional threat circuit. They convincingly show that regions beyond the threat circuit significantly contribute to processing of threat and safety, and that different regions tend to be associated with threat, safety and adaptive switching between the two. Importantly, the authors present results from a discovery as well as two validation data sets, which is a great practice towards generalizable science. The methodology used is robust, though I have some concerns and questions which I present below.*

Response: We are immensely grateful for the positive feedback, and sincerely appreciate the insightful comments and suggestions. We have made a sincere effort to carefully address your requests, suggestions, and concerns point-to-point as detailed below.

Comment 1: *“A first-order autoregressive model was used to account for the temporal structure of the noise” a) What time points were the classifiers trained on? b) How did the authors ensure that the outcome from the previous trial was not included in the decoded signal? c) The AR(1) model is known to perform poorly at removing autocorrelation (Mumford et al. 2012; <https://doi.org/10.1016/j.neuroimage.2011.08.076>), this is particularly important at dissociating the actual outcomes (shocks, pain etc) from threat. The authors should either demonstrate lack of such signals or take thorough steps to remove the outcome signal.*

Response: For questions **a** and **b**: we used activation maps (beta maps) estimated from GLM rather than extracting time points from each trial as features to train the classifiers. Specifically, we divided each experimental phase into 4 trial-blocks, and modeled each CS type in each trial-block using a regressor in the GLM. Thus, for each subject, we obtained two CS beta maps for each trial-block (one for CS+, the other for CS-), and used them for classification. This analysis strategy limited the signal from previous trial carry over to the next trial, which we hope addresses the concern raised above. We added the following sentence to the main text to clarify that we used activation maps as features for the classification.

“The activation maps of each CS type in each TB were estimated via the general linear model (GLM), and separately used to build a decoding model for each TB.”

For question **c**, we believe that the shock response was not likely to have a major impact on the signal for the classification for two main reasons. First, there were no shocks delivered in extinction learning and recall phases in the experimental paradigm, and we still obtained high accuracies in the first trial-blocks of these phases. Second, the shock response was modeled in GLM as a regressor for the conditioning phase to remove its impact. Nonetheless, to further examine this issue and make sure we fully address the

reviewer's concern, we conducted a new analysis in which we only including CS+ trials that were not reinforced in the conditioning phase, thereby fully excluding the impact of shock on CS+ trials. The results of these analyses, shown in the figure below, revealed comparable results to those reported in the manuscript, giving us much more confidence that our results are not robustly impacted by shock delivery during the conditioning phase. We added the following sentence to clarify it.

"In the conditioning phase, the shock response was not likely to have a major impact on the classification performance, since we obtained comparable results in decoding unreinforced CS+ from CS- (Supplementary Fig. 2)."

Supplementary Figure 2. Classification in the conditioning phase by only including unreinforced CS+ trials. Classification performance based on 'threat circuit' patterns on the discovery dataset (A), validation dataset 1 (B), and validation dataset 2 (C). Classification performance based on 'whole-brain' patterns on the discovery dataset (D), validation dataset 1 (E), and validation dataset 2 (F). The unreinforced CS+ trials were divided into two trial-blocks (3 CS+ trials in each trial-block). Activation maps for unreinforced CS+ trials and corresponding CS- trials within each trial-block were estimated and used for classification analysis.

Comment 2: "Considering the dynamic nature of the threat/extinction learning processes, we divided each experimental phase into 4 trial-blocks (TB, from TB1 to TB4) and built a decoding model for each trial-block separately." The number of trials that went to each trial block should be clarified in the introduction/results - is it always 4? Are there any

differences between studies in the discovery dataset and/or between discovery and validation datasets?

Response: The reviewer is correct that each trial-block contained 4 trials of each CS type (i.e., 4 CS+, 4 CS-). And the discovery dataset and the validation datasets 1-2 used the same trial-block definition because the three datasets used the same paradigm design from Milad lab. We made the following revisions in the manuscript to clarify this matter:

“... we divided each experimental phase into 4 trial-blocks (TB, from TB1 to TB4). Since each experimental phase contained 32 trials in total, each TB contained 8 trials (4 trials for each CS type). The activation maps of each CS type in each TB were estimated via the general linear model (GLM), and separately used to build a decoding model for each TB.”

“Next, we tested the generalizability of the decoding models on two external validation datasets (N = 98 and N = 127, respectively) in which participants underwent the same paradigm as those in the discovery dataset^{35,36}. The paradigm setups (e.g., trial number, images used as CS) of the two external datasets were the same as in the discovery dataset.”

Comment 3: *CS+E and CS+U in extinction seem to both drop off, however, I would expect that CS+U would remain higher since it remained threatening during extinction. How do the authors interpret this, especially since in the discovery set and in validation set 1 CS+E seems to be higher than CS+U (in threat circuit analysis).*

Response: This is a great question. We believe the reason for the drop off in classification accuracy for the CS+U is additional extinction training occurring during extinction recall—much like what is observed during extinction learning phase. We remind the reviewer that we do not deliver any more shocks during the extinction recall test. It is well documented in the human extinction literature that, unlike extinction in rodents, humans extinguish exceptionally quick in these de novo conditioning and extinction paradigms—typically within 2 trials. One prospective study to test this idea would be to re-condition one of the CS+s (pair with more shocks) during the recall phase and see if this eliminates the drop. We acknowledge this drop and its likely explanation in the Discussion section of the manuscript as follows.

“Similarly, during early extinction, the distinction between the CS+ which represents threat, and the CS- which represents safety, is diminished due to extinction learning within a few trials. The diminished difference between CS+ and CS- led to rapidly decreased decoding accuracy after early extinction which ended up in chance-level performances in late extinction (Fig. 1A-C, 2A-C). The same pattern of drop in accuracies was also observed during extinction memory recall. This is the case for the extinguished (CS+E) and unextinguished (CS+U) cues. The rapid decrease in accuracies is consistent with the well-documented exceptionally quick extinction learning in humans both during extinction learning and extinction memory recall^{3,66}.”

Comment 4: *It is a bit unclear how the data sets were selected - are these all available published data sets using the Milad et al. paradigm? If not, what was the selection process?*

Response: For the discovery dataset, we used all available data collected at Massachusetts General Hospital (MGH) by Milad laboratory that used the Milad et al. paradigm to maximize the sample size. The external validation datasets 1-2 were from two other studies that used the same paradigm as the discovery dataset. The data for those 2 datasets were acquired by colleagues of Dr. Milad at MGH. All data that were provided to us by our colleagues were included in the external validation datasets. We

added the following sentence to the manuscript to clarify this point. We also added a Supplementary table that listed the information of each included study.

“This discovery dataset was constructed by all available data from our laboratory that used the Milad et al. paradigm^{25,26} to maximize the sample size (Supplementary Table 1).”

Comment 5: *In Figure 2D, why does extinction only have 1-4 CS? The number of trials is the same as acquisition (p16).*

Response: The reviewer is correct that the whole extinction phase contained 16 trials of each CS type as in conditioning. We showed the predictive pattern of 1-4 CS (i.e., the first trial-block) in Fig. 2D only because the other trial-blocks did not result in robust classification performance (Fig. 2A-C). Similarly, we only showed the predictive pattern of 1-4 CS+E/CS+U in the recall phase. We clarified this in the revised manuscript by adding the following sentence:

“). For extinction learning and extinction memory recall phases, we only show the predictive patterns of the first trial-blocks because these were the trial-blocks that showed the most robust classification performance in the two phases.”

We think using 1-4 CS to represent the first trial-block in the figures was a little confusing; we now modified the figures to use ‘Trial-block 1’ instead.

Comment 6: *It only becomes clear in discussion that the discovery dataset uses exclusively Milad et al. task - add to introduction/results.*

Response: Thank you very much for this comment- we missed it. We added the following sentence into the Results section according to the reviewer’s suggestion.

“This discovery dataset was constructed by all available data from our laboratory that used the Milad et al. paradigm^{25,26} to maximize the sample size (Supplementary Table 1).”

Comment 7: *“The optimal hyperparameter was selected based only on training data” Please share the results of the L2-regularization tuning, including the final value used in all analyses.*

Response: We apologize that we did not describe the hyperparameter selection procedure well. The hyperparameter (i.e., L2-regularization term) was selected automatically (using the GridSearchCV function from Scikit-Learn) based on training data within the 5-fold cross-validation procedure (i.e., 4-folds of data) during each iteration. Thus, there were 5 different values of the hyperparameter (one for each of the 5 cross-validation iteration). And since we repeated the cross-validation procedure 20 times using different dataset splits (randomized), there are a total of 100 (5*20) values for each time-block. Across all time-blocks, there would be 1200 (12 time-blocks * 100 values) number of tuned values. Thus, it is hard to visualize the hyperparameter tuning results and report the final values for the cross-validation analysis. However, for the generalization analysis (training on discovery dataset, testing on validation datasets 1-2), since we trained the models based on all data from the discovery dataset, there were only two models (one for the ‘threat circuit’ classifier, the other for the ‘whole-brain’ classifier) for each time-block, we share the tuned values of the hyperparameter on GitHub: <https://github.com/zhenfuwen01/threat-mvpa..>

Comment 8: *(introduction) “Past research within this field has primarily focused on the roles of a few subcortical and cortical structures in how the association between the conditioned and the unconditioned stimuli is formed, and how the defensive responses are generated and subsequently extinguished.” Perhaps since authors are referring to the past research into Pavlovian threat reactions, they should also refer the reader to the Fullana (2016) article <https://www.nature.com/articles/mp201588>.*

Response: We now add the Fullana et al. 2016 article as another reference in the Introduction as recommended.

Comment 9: *(introduction) “For example, the amygdala is a key node in the acquisition and extinction of conditioned threat; and the hippocampus is critical for contextual information processing” Please cite.*

Response: We added the references in the revised manuscript.

Comment 10: *Please share analysis code together with reproducibility instructions, as well as the resulting safety/threat maps.*

Response: We uploaded our analysis codes, brain masks, resulting brain maps, and instructions to GitHub: <https://github.com/zhenfu-wen01/threat-mvpa>. We also added the Data availability and Code availability sections into the revised manuscript.

Comment 11: *Reference 7 contains full first name.*

Response: We carefully checked all the references and corrected them in the updated manuscript.

Reviewer #2

General comment: *Despite its basic and translational importance, the neurobiology of fear remains incompletely understood, impeding the development of more effective interventions for maladaptive fear. Most human imaging studies employ small samples and conventional mass univariate analyses, which have important inferential limitations. Thus, the focus of this paper is timely and likely to be of interest to a broad spectrum of researchers and clinical scientists.*

Key Strengths

- *extraordinarily large sample for this domain of research, with the potential for enhanced power, precision, and reproducibility.*
- *fairly comprehensive tests of sensitivity, specificity, and generalizability via multiple independent hold-out datasets (i.e. not used for classifier training).*
- *the innovative MVPA classifier approach has the potential to address questions of specificity that cannot be addressed using traditional mass univariate approaches, innovation is strengthened by the thoughtful bin-wise approach, which provides important clues about temporal dynamics.*
- *the first author (Dr. Wen) is to be congratulated on what amounts to a truly enormous amount of work (e.g. data wrangling, processing, and analyses for 9 datasets)*

Key Weaknesses

- *the Introduction is unclear and occasionally (mildly) misleading*
- *Some of the claims in the Discussion go beyond the data and analyses*

In the section that follows, I provide a few suggestions for further strengthening the manuscript.

Response: We are very grateful for the positive feedback. We sincerely appreciate the reviewer's insightful comments and suggestions. We have carefully addressed your requests, suggestions, and concerns point-to-point, as detailed below.

MAJOR / GENERAL

1.INTRODUCTION.

Comment 1: *The study (while useful, significant, and innovative) is poorly motivated in the introduction. e.g.*

"Although largely supported by cross-species research, this simplified model based on Pavlovian reactions does not fully account for the range of complex cognitive and memory representations that are formed and stored in distributed neural circuits and used to guide deliberative defensive actions. While the 'threat circuit' is highly necessary for the detection and response to threat, there is a need for studying how this circuit interacts with other cognitive and sensory processing neural networks^{10,11}. Currently, we do not have established distributed neural decoders that are integrative, sensitive, and reproducible for the detection and response to threat and safety. To move beyond simplified circuits of threat detection and reaction, we examined the complex neural circuits predictive of Pavlovian threat learning throughout the human brain to reveal a more integrated and updated conceptualization that highlights the dynamic interplay between multiple neural systems to detect the threat, and to form implicit and explicit memory representations for prospective defensive actions."

Nothing about "circuits," "networks," "how...[the threat-anticipation] circuit interacts with other...networks," or "the dynamic interplay between multiple neural systems" motivates the development of a brain signature based on sluggish BOLD responses to Pavlovian cues. There are excellent reasons for using MVPA, but these are not them.

[Note: This concern remains true even when considering the block-wise RSA analyses]

Response: We agree with the reviewer that the motivation of using MVPA was not well described. We reorganized the Introduction section as follows so that we explicitly demonstrate the reasons for using MVPA.

“Currently, the field lacks established distributed neural representations that are sensitive and reproducible for the encoding of threat and safety cues across stages of threat conditioning and its subsequent extinction learning and memory retrieval. A compelling approach to study this is the machine learning-based multivariate pattern analysis (MVPA)¹⁶. In contrast with the traditional univariate analysis examining localized activation, MVPA enables the integration of neural patterns across distributed regions to sensitively detect subtle differences between mental processes. The cross-validation and external validation procedures of MVPA provide ways to examine the generalizability of activation patterns. Considerable evidence has shown that the MVPA is more stable and sensitive than traditional univariate analysis^{16–18}. In addition, MVPA could examine whether neural representations are similar across conditions or processes even for overlapping univariate activations¹⁹, and thus allows one to test the specificity of observed patterns to conditioned threat compared to other triggers of negative affect.”

Comment 2: *A key innovation of the approach is the focus on extinction (day 1) and extinction recall (day 2), but this appears "out of the blue" several hundred words into the introduction. I would encourage the authors to mention/foreshadow this earlier. This is particularly important in light of evidence suggesting that extinction is likely more relevant to pathological fear and anxiety.*

Response: We appreciate the reviewer's thoughtful suggestion. In light of this and other comments by the reviewer regarding the introduction, we have now revised the Introduction to take into consideration the mention of extinction learning and recall earlier in the Introduction.

Comment 3: *Based on the introduction, it's not clear what the 'second analytic approach' entails. e.g.*

"we investigated the contribution of distributed neural systems in the formation of memory representations at higher cortical levels, which may be related to encoding deliberative defensive actions."

As a consequence, it's not clear what is meant by the next section:

"From the combined first and second analytic approaches, we constructed an extended circuit that includes sensory, working memory, and cognitive neural nodes. We trained new classifiers based on neural activations from this extended circuit obtained from our threat conditioning datasets,"

[are the authors referring to the 'virtual lesion' approach described in 'Decoding based on w-b patterns?' ... cf. Fig. S1b]

Response: We agree with the reviewer that the two analytic approaches were not clearly described. The two analytic approaches to which we referred are: 1) decoding based on the 'threat circuit' activations; 2) decoding based on the 'whole-brain' (excluding the 'threat circuit') activations (i.e., the 'virtual lesion' approach). We revised the paragraph to better convey the information as follows.

"In the first approach, we examined fMRI-based neural patterns within the 'threat circuit' in decoding stimuli that have been conditioned to signal threat from those that have been conditioned to signal safety. In the second approach, we examined neural patterns beyond the 'threat circuit' in contributing to conditioned threat and safety representations. Specifically, based on recent conceptualizations that more distributed neural systems are engaged in threat processing^{14,15,20-24}, we conducted decoding analysis using distributed 'whole-brain' activation patterns (with the 'threat circuit' excluded). ... By combining the 'threat circuit' and the brain regions identified in the second analytic approach, ..."

Comment 4: *Minor/Moderate - A number of minor claims made "in passing" are potentially misleading to readers who are either unfamiliar with this literature, or who are only familiar with the animal or imaging arms of it. See my minor/specific comments, below.*

Response: We revised the manuscript according to the reviewer's comments. We made every attempt possible to be very careful not making any statements that might be considered potentially misleading.

Comment 5: *Some of the terminology/nomenclature drifts across the paper. Please revise for consistency.*

Response: We carefully revised the manuscript to increase consistency.

2. METHOD.

Comment 6: *Participants - Authors should add a table to the main report or Supplement with basic demographic and scanning information for the studies that contributed to the 1,465. Authors should clearly indicate whether each study was used for training/discovery vs. hold-out testing. Something similar should be done, at least in the Suppl, for the "external datasets." Here, the authors should briefly indicate the nature of the paradigm employed in each external dataset.*

Response: A wonderful suggestion and immensely helpful; thank you. We added two tables into the Supplement according to the reviewer's suggestion, as listed below.

Supplementary Table 1. Basic information of discovery dataset and validation datasets 1-2.

Dataset	Training or testing	Demographic information	Key scanning information	Paradigm description	Original publications
Discovery	Training	Sample size: 31 (45% female); Age (SD): 32.1 (11.5); Diagnosis: 16 PTSD, 15 TENC.	Setting 1: Siemens 3T MRI scanner, 8-channel head-coil. Functional data TR: 3.0 s, slice number: 45, voxel size: 3 × 3 × 3 mm.	Threat conditioning and extinction paradigm, using colors as CS (duration: 6 s), 62.5% reinforcement rate. Threat conditioning: 16 CS+, 16 CS-; Extinction learning: 16 CS+, 16 CS-; Extinction memory recall: 8 CS+E, 8 CS+U, 16 CS-.	Milad et al. 2009 ¹ .
Discovery	Training	Sample size: 34 (100%); Age: 23.2 (2.6); Diagnosis: 34 HC.	Same as setting 1.	Same as above.	Zeidan et al. 2011 ² .
Discovery	Training	Sample size: 57 (41%); Age: 27.9 (8.2); Diagnosis: 57 HC.	Same as setting 1.	Same as above.	Milad et al. 2007 ³ , Milad et al. 2013 ⁴ , Holt et al. 2012 ⁵ .
Discovery	Training	Sample size: 65 (54%); Age: 34.0(13.2); Diagnosis: 21 HC, 24 PTSD, 20 TENC.	Setting 2; Siemens 3T MRI scanner, 32-channel head-coil. Functional data TR: 2.56 s, slice number: 48, voxel size: 3 × 3 × 3 mm.	Same as above.	Marin et al. 2016 ⁶ .
Discovery	Training	Sample size: 114 (61%); Age: 30.5 (12.0); Diagnosis: 21 HC, 93 ANX.	Same as setting 2.	Same as above.	Marin et al. 2020 ⁷ .
Discovery	Training	Sample size: 124 (76.6%); Age: 38.6 (17.5); Diagnosis: 125 HC	Same as setting 2.	Same as above.	Unpublished.
Validation dataset 1	Testing	Sample size: 98 (64.3%); Age: 32.0 (8.4); Diagnosis: 98 HC.	Setting 3. Siemens 3T MRI scanner, 32-channel head-coil. Functional data TR: 3.0 s, slice number: 48, voxel size: 2.5 × 2.5 × 2.5 mm.	Same as above.	Sevinc et al. 2019 ⁸ .
Validation dataset 2	Testing	Sample size: 127 (68.5%); Age: 24.6 (5.0); Diagnosis: 63 PTSD, 64 TENC.	Same as setting 2.	Same as above.	Seo et al. 2021 ⁹ .

PTSD: Post-traumatic stress disorder; TENC: Trauma-exposed non-PTSD control; ANX:

Anxiety disorder; HC: Healthy control; SD: Standard deviation.

Supplementary Table 3. Basic information of external validation datasets 3-9.

Dataset	Training or testing	Demographic information	Key scanning information	Paradigm description	Original publications
Validation dataset 3	Testing	Sample size: 299 (58.6% female); Age (SD): 33.9 (10.1); Diagnosis: 299 HC	GE 3T MRI scanner, 8-channel head-coil. Functional data TR: 2400 ms, slice number: 47, voxel size: 3.0 × 3.0 × 3.0 mm.	Visual conditioning task, using two male three-dimensional virtual humanoid characters as CS, 16 trials per CS, 50% reinforcement rate. Stimulus duration: 6 s.	Vinberg et al. 2022
Validation dataset 4	Testing	Sample size: 94 (69.4%); Age: 22.1 (3.3); Diagnosis: 94 HC	Philips 3T MRI scanner, 8- or 32-channel head-coil. Functional data TR: 2000 ms, slice number: 38 or 39, voxel size: 2.4 × 2.4 × 3.1 mm, or 3 × 3 × 3.3 mm.	Visual conditioning task, using face and house images as CS, 13 trials per CS, 46.2% reinforcement rate. Stimulus duration: 6 s.	Visser et al. 2021
Validation dataset 5	Testing	Sample size: 48 (66.7%); Age: 23.5; Diagnosis: 24 PTSD, 24 TENC	Siemens 3T MRI scanner, 32-channel head-coil. Functional data TR: 2000 ms, slice number: 48, voxel size: 3 × 3 × 3 mm.	Visual conditioning task, using two categories of objects as CS, 24 trials per CS, 50% reinforcement rate. Stimulus duration: 4.5 ± 0.5 s.	Henning et al. 2021
Validation dataset 6	Testing	Sample size: 68 (66.2%); Age: 29.6 (15.9); Diagnosis: 68 HC	Siemens 3T MRI scanner, 32-channel head-coil. Functional data TR: 2000 ms, slice number: 39, voxel size: 3 × 3 × 3 mm.	Auditory conditioning task, using two tones as CS, 16 trials per CS, 50% reinforcement rate. Stimulus duration: 4 s.	Reddan et al. 2018
Validation dataset 7	Testing	Sample size: 65 (50.7%); Age: 21.5 (2.1); Diagnosis: 65 HC	GE 3T MRI scanner. Functional data TR: 2000 ms, slice number: 36, voxel size: 3.125 × 3.125 × 3.8 mm.	Task assessed 'subjective fear' to intrinsically salient images. Participants were presented with 80 different pictures and were instructed to report the fearful state they experienced for the stimuli. Stimulus duration: 15 s.	Zhou et al. 2021
Validation dataset 8	Testing	Sample size: 182 (52.0%); Age: 42.8 (7.3); Diagnosis: 182 HC	Siemens 3T MRI scanner, 12-channel head-coil. Functional data TR: 3000 ms, slice number: 34, voxel size: 3.125 × 3.125 × 3 mm.	Task examined negative affect induced by intrinsically salient images. Participants were presented with 15 neutral and 15 negative pictures and were instructed to report their emotional state. Stimulus duration: 7 s.	Chang et al. 2015
Validation dataset 9	testing	Sample size: 59 (51.7%); Age: 20.8 (3.0); Diagnosis: 59 HC	Siemens 3T MRI scanner, 12-channel head-coil. Functional data TR: 2000 ms, slice number: 33,	Tasks examined perception of physical pain or social pain. Participants were presented with heat/warm thermal stimuli	Woo et al. 2014

			voxel size: 3.75 × 3.75 × 3 mm.	or images of ex-partner/close friend. Stimulus duration: 15 s.	
--	--	--	---------------------------------	---	--

PTSD: Post-traumatic stress disorder; TENC: Trauma-exposed non-PTSD control; HC: Healthy control; SD: Standard deviation.

Comment 7: *There is drift in the terminology/nomenclature. Study vs Setting. The Method section does not use the First/Second Analytic Approach terminology. Please revise for consistency across the entire paper. Moving some of the imaging parameter details to tables (see above) might make this easier for readers to digest.*

Response: We agree with the reviewer and revised the Method section accordingly to match up with the main text.

Comment 8: *There are longstanding concerns that the Juelich/SPM amygdala masks are poorly aligned with the probabilistic MNI152 templates (e.g. Tillman et al. Hum Brain Mapp 2018 Figure S5). It is not clear based on Fig S1a whether this is a concern here. Authors should provide a coronal slice in the supplement to enable readers to judge for themselves. In the event that inspection reveals inadequate alignment, authors should appropriately modify the approach.*

Response: We appreciate the reviewer's suggestion. Based on our visual inspection, the amygdala masks we used here appear to well align with the MNI152 template. We added a coronal slice of amygdala in the Supplement for the reviewer and readers to judge the alignment. We also added a sentence in the Methods section to clarify it.

“There are concerns that the Juelich/SPM amygdala masks are poorly aligned with the probabilistic MNI152 templates (e.g.⁷³). Based on our visual inspection, the amygdala masks we used here were well aligned with the MNI152 template. We added a coronal view of the amygdala masks (Supplementary Fig. 9) to enable readers to judge the alignment.”

Supplementary Figure 9. **The map of amygdala masks.** BLA: Basolateral amygdala; CMA: Centromedial amygdala.

Comment 9: *Reducing vmPFC, sgACC, and MCC/dACC to 8-mm spheres (cf. Fig S1a) is an unexpected choice; this means that the vast majority of voxels in those 3 cortical*

regions cannot contribute to the classifier. It also means that some of the "other" regions (e.g. hipp, alns) will actually contribute more measurements to the classifier training than these ostensibly "larger" cortical regions. Furthermore, it also means that many, many vmPFC and MCC/dACC voxels that fall outside of the a priori masks will contribute to the "whole-brain" classifier, with potentially significant implications for interpretation (i.e. the Discussion is framed around broad regions, such as "dACC"). I do not think this curious choice is inherently problematic, but I would encourage the authors to appropriately motivate this choice in the Method section for curious readers.

Response: We agree with the reviewer that many voxels in the vmPFC, sgACC and MCC/dACC could not be fully covered by the 8 mm spheres. We decided to use 8 mm spheres around the meta-analysis defined coordinates to define the ROIs because these voxels are most likely to be involved in conditioning and extinction (based on meta-analysis). It is possible that voxels that fall outside of the a priori masks contributed to the "whole-brain" classifier, but these contributions would be very limited because 1) as described in the Methods, "To eliminate the impact of the 'threat circuit' on the decoding, voxels from the 'threat circuit', and their neighborhoods within a radius of 3 voxels, were excluded from the analysis." Since each voxel in the preprocessed images was 2×2×2 mm, thus 14 mm spheres were used to exclude voxels from the 3 regions in the analysis, which limited the contributions of the remaining voxels from the 3 regions to the 'whole-brain' classifiers. 2) from Fig. 2D and Supplemental Fig. 5, we can see that the predictive patterns of the 'whole-brain' classifiers were not heavily distributed near the 3 regions.

To make sure we fully address and satisfy the reviewer's concern, and further assure that the choice of ROI definitions would not largely impact the results, we conducted an additional analysis by excluding voxels from the 'whole-brain' classifiers in a more aggressive way. Specifically, we excluded the anterior cingulate cortex, paracingulate cortex, and medial prefrontal cortex dorsal to Z = 0, together with other nodes of the 'threat circuit' from the 'whole-brain' activation patterns, and conducted the classification analyses. The classification performances were comparable to those in Figures 2A-2C, suggesting that our choice in defining vmPFC, sgACC, and dACC would not largely impact the obtained results. We included the analysis in the Supplementary Materials and acknowledged the choice of masks as a limitation in the main text.

"Several limitations of this study should be noted. First, the masks for some 'threat circuit' regions, including vmPFC, sgACC, and dACC, were defined using small spheres around coordinates derived by meta-analysis. These masks could not fully cover the three relatively large cingulate cortices. Thus, patterns of activations within voxels outside of these masks may have contributed to the 'whole-brain' classification. We note that the size of the mask definition is unlikely to have impacted the 'whole-brain' classification results. This is because we conducted additional analyses using much larger masks (thus excluding more voxels from the same cingulate regions) and the outcome of the 'whole-brain' classifiers did not change (Supplementary Fig. 8)."

Supplementary Figure 8. Classification with more voxels from the ‘threat circuit’ excluded. **A.** Cross-validation accuracies on the discovery dataset. **B.** Generalization accuracies on the validation dataset 1. **C.** Generalization accuracies on the validation dataset 2. Instead of defining vmPFC, dACC, sgACC using 8 mm spheres, we used a more inclusive mask to define these regions. The mask includes the anterior cingulate cortex, paracingulate cortex, and a larger part of vmPFC. The vmPFC was defined by including frontal pole, frontal medial cortex, subcallosal cortex, paracingulate gyrus, anterior cingulate gyrus and frontal orbital cortex (areas dorsal to $Z = 0$ mm and lateral to $X = \pm 12$ mm are excluded). All regions were defined based on the Harvard-Oxford atlas. We excluded voxels from this inclusive mask and the other nodes of the ‘threat circuit’, and then conducted the ‘whole-brain’ classification analysis.

3. ANALYTIC STRATEGY.

4. RESULTS.

Comment 10: *The Haufe-transformed weight shown in fig 1d are presented in such a superficial way in the main text that they add little to the paper. These are really quite interesting! Authors are strongly encouraged to adequately describe them in the main text - walk the reader thru which regions prefer threat, which prefer safety, and which shown bin-dependent “dynamics” (but see my comments about the Discussion, below).*

Response: We agree with the reviewer on this. We now added the following sentences to describe the predictive patterns in the main text.

“This analysis showed several interesting results. First, subregions of the amygdala and insular cortex differentially contributed to decoding the CS+ and CS- across trial-blocks. For example, during early conditioning and recall, the BLA mostly exhibited predictive patterns towards the CS-, while the CMA exhibited predictive patterns associated with the presentation of the CS+, only during the first trial-block of conditioning, which then subsequently switched to encoding for the CS-. Second, several regions exhibited consistent predictive patterns across decoding models of different trial-blocks. Specifically, dACC and dAI consistently coded CS+, while the hippocampus and PI mainly coded CS-. Third, the predictive patterns of regions such as amygdala, sgACC, and vmPFC were adaptive- changing the coding signal between the CS+ and CS- across trial-blocks and phases, suggesting that these regions were dynamically involved across the experiment. It is important to note that we did not conduct significance tests on the changes of encoding across trial-blocks within each of these subregions, and that these changes may reflect those similar to a habituation effect.”

Comment 11: *The virtual lesion results are interesting, but the verbal framing seems to*

muddle 2 ideas: (a) maximizing predictive performance using more voxels versus (b) the degree to which the 'threat-anticipation circuit' (ie the usual suspects) is critical for decoding performance. (This is also something of an issue in the Discussion; see below).

"Although significant, the decoding accuracies based on the 'threat circuit' activation patterns were often below 75%. Given the conceptual framing stated above, we reasoned that considering the neural representations from multiple sensory, attention, and cognitive circuits should improve the discrimination accuracy of the classifier. Indeed, using neural representations from the entire brain (excluding the 'threat circuit') led to substantially improved decoding performances (Figure 2A)."

I would encourage the authors to refine the framing to sidestep muddling. They *might* also wish to add a supplementary analysis that uses *all* GM voxels (i.e. the logical-AND conjunction of the voxels used in figure 1 and figure 2), as this would answer an obvious question many readers will have.

Response: We totally agree with the reviewer that the motivation of the classification analysis using brain regions outside the 'threat circuit' (i.e., the virtual lesion analysis) was not clearly presented. As is well-summarized by the reviewer, the purpose of the virtual lesion analysis is to examine the degree to which the neural systems outside the 'threat circuit' support the conditioned threat decoding, rather than to maximize predictive performance by including more voxels. We revised the sentences below to make the purpose of the virtual lesion analysis clearer.

"The decoding performances based on the 'threat circuit' activation patterns are encouraging, and support the well-documented contributions of these neural nodes to associative learning during threat conditioning and extinction learning. Yet as noted in the introduction, there are several prior studies now pointing out the significant contributions of multiple other brain regions to threat and safety signaling. We therefore moved forward to examine how activations from multiple sensory, attention, and cognitive circuits might also contribute to the encoding of CS+ and CS- information across learning phases."

Furthermore, we also followed the reviewer's suggestion to conduct the supplementary analysis that used all gray matter voxels (including the 'threat circuit' voxels) for the classification. The results of this analysis were included in the Supplementary Materials in the revised manuscript and referred to in the main text as follows.

"For completeness, we conducted additional analysis by using all gray matter voxels (including the 'threat circuit' voxels) for the classification, which resulted in similar performance as those in Fig. 2A (Supplementary Fig. 4)."

Supplementary Figure 4. Classification based on all gray matter patterns. A. Cross-validation accuracies on the discovery dataset. **B.** Generalization accuracies on the validation dataset 1. **C.** Generalization accuracies on the validation dataset 2. Activation patterns of all gray matter voxels (including the ‘threat circuit’ voxels) were used for the classification.

Comment 12: *Figure 3a seems deeply misleading. See my prior comment about using 8-mm spheres to sample vmPFC, dACC/MCC, etc. Please revise to make it clear that some regions are reduced to small spheres. Also - it is difficult to see the division between CMA vs. BLA ... please use a darker/bolder border as with the other regions (or the broken line used in vmPFC-vs-sgACC).*

Response: Our intention of generating this figure was to provide the reader with a schematic illustration of the extended network and not in any way to provide a 100% precise representation of each ROI and node we examined and discussed. But we very much appreciate the comment from the reviewer. With the reviewer’s well-taken point in mind, we modified Figure 3a according to the reviewer’s suggestions to make it clear that some regions are reduced to small spheres. But we also state in the manuscript and the figure legend that this is a schematic representation. Please see below for the revised Fig. 3A.

Figure 3. The ‘threat detection and flexible responding circuit’ and its generalizations to external datasets. A. Schematic illustrative map of the extended ‘threat detection and flexible responding circuit’. Red-colored regions consistently code the CS+ across experimental phases. Blue-colored regions consistently code the CS- across experimental phases. Yellow-colored regions dynamically code CS+ or CS- depending on the experimental phase.

Comment 13: *"Based on the regions that exhibited high contribution to the decoding models, and the traditional ‘threat circuit’, we constructed an ‘extended threat detection and responding circuit’ (Figure 3A), which, based on our data, better defines the brain regions involved in threat processing than the ‘threat circuit’ alone." Seems odd to jump to RSA, before describing performance for the "extended" classifier/decoder.*

Response: The logic that we described the RSA results before the classification results is that: we start with identifying the regions that exhibited high contribution to the decoding models; we then examine how these regions contributed to the decoding (i.e., they coded for threat, safety, or exhibited dynamic coding) using RSA. After characterized the roles of these regions in decoding, we then test the generalizability of classifiers trained using

activations of the extended circuit. Therefore, if acceptable to the reviewer, we prefer to keep the order as it is. However, if the reviewer feels strongly about reordering the RSA and external validation results, we would be delighted to comply.

Comment 14: *"The first community (red color in Figure 3A) consistently coded for threat (CS+) across conditioning, extinction learning, and recall. The second community (blue color in Figure 3A) consistently coded the safety of the cue (CS-). The third community (yellow color in Figure 3A) showed dynamic coding; shifting their signal between threat and safety (CS+ or CS-) depending on the experimental phase."*

Please summarize the regions in each community in the main text - thanks!

Response: We appreciate the reviewer's suggestion. In the revised manuscript, we first clarified region names of the 'threat circuit' and the 14 newly identified brain regions. We then summarize the regions in each community.

"First, we used activation patterns of the 'threat circuit', which mainly included basolateral and centromedial amygdala (BLA, CMA), anterior and posterior parts of hippocampus (aHPC, pHPC), subregions of the insular cortex (dorsal anterior part, dAI; ventral anterior part, vAI; posterior part, PI), dorsal anterior cingulate cortex (dACC), subgenual ACC (sgACC), and ventromedial prefrontal cortex (vmPFC), for the decoding analysis."

"Based on these predictive patterns, we identified 14 representative brain regions that significantly contributed to the decoding across multiple trial-blocks (Fig. 3A), including: (1) angular gyrus (AG), (2) orbital frontal cortex (OFC), (3) supplementary motor area (SMA), (4) primary somatosensory cortex (S1), (5) primary motor cortex (M1), (6) visual cortex (VIS)/Occipital pole, (7) cerebellum (CER)/Crus I, (8) thalamus (TH), (9) opercular part of the inferior frontal cortex (Opr), (10) triangular part of inferior frontal gyrus (Tri), (11) middle frontal gyrus (MFG), (12) caudate nucleus (Cd), (13) superior frontal gyrus (SFG), (14) posterior cingulate cortex (PCC). The results of identified regions were robust to criteria in defining the percentage of voxels included for the analysis (Methods, Supplementary Fig. 6)."

"The first community (red color in Fig. 3A) consistently coded for threat (CS+) across conditioning, extinction learning, and recall, which included the dACC, dAI, PCC, Opr, Cd, and TH. The second community (blue color in Fig. 3A) consistently coded the safety of the cue (CS-), which included the the PI, aHPC, pHPC, and OFC. The third community (yellow color in Fig. 3A) showed dynamic coding; shifting their signal between threat and safety (CS+ or CS-) depending on the experimental phase. This community included the BLA, CMA, vAI, vmPFC, sgACC, AG, S1, M1, SMA, MFG, VIS, Tri, and CER."

5. DISCUSSION / IMPLICATIONS.

Comment 15: *Some key interpretive claims need to be unpacked. For instance, one of the major take-home points is that the present results reinforce theoretical work underscoring the importance of regions beyond 'the usual suspects' for processing threat, but this is simply claimed and tagged with a list of numeric references. It needs to be briefly unpacked, and linked to the theoretical literature. See my minor/specific comments below for details.*

Response: Please see our response to the reviewer's comments below.

Comment 16: *A major take-home point is that the decoders work quite nicely even when*

you "virtually lesion" all of the usual suspects, but this is barely mentioned at all in the Discussion.

Response: We appreciate the reviewer's comment. In the original submission, we tried to emphasize this as we do agree that this is a critical point. We now made additional modifications to the Discussion section to emphasize this point. For example, we added the following sentence to the first paragraph of the Discussion section:

"Importantly, however, the classification accuracies were numerically superior when activation patterns from other distributed neural systems (excluding the 'threat circuit') were used."

We also reorganized the Discussion section, and explicitly discussed this topic in the third paragraph of this section.

Comment 17: *Authors need to be more precise in their language. Reliable, critical, identify/detect, and respond have specific meanings that do not correspond to how they're used in the paper. See my minor/specific comments below for details.*

Response: We revised the manuscript thoroughly to be more precise in the language we use.

Comment 18: *"In contrast, multiple brain regions, e.g., the dorsal anterior insular cortex and the dorsal anterior cingulate cortex (dACC) consistently contributed to the classification of the CS+, which support the role of these regions in the encoding of the associative aspect related to threatening cues and/or to the generation of conditioned threat responding."*

*This interpretative claim goes well beyond that licensed by the data and approach. There is nothing in the results shown in fig 1d and 3a that rules out the possibility that these specific regions are equally (or more) responsive to intrinsically/unlearned aversive stimuli [i.e., the tests are whole-decoder, not region specific]. And it is certainly the case that these regions respond (in a mass univariate sense) to a broad range of stimuli (cf. meta-analyses by De La Vega...& Wager/Yarkoni; Chang...& Yarkoni; Shackman...& Davidson; and work by Uddin/Pessoa). Please fix the claims to align with the results, or vice versa. The authors *might* consider adding some supplementary analyses (e.g. x-validated x-classification) to determine whether these specific regions are preferentially/differentially associated with pavlovian threats, but this is not essential.*

Response: We agree with the reviewer that the results reported in the manuscript could not answer the question that whether these regions are preferentially associated with Pavlovian threats. We revised the sentences to clarify it.

"Of note, the brain regions discussed above also respond to a broad range of stimuli⁵⁹⁻⁶², and our experimental design and analyses used in our study could not determine whether they are preferentially associated with Pavlovian threats vs. other processes."

Comment 19: *"On the other hand, several neural nodes showed transient preferences in the encoding of the CS+ and CS- across experimental phases...like the MFG, SFG, vmPFC, and amygdala, amongst others."*

While the CeMe, BLA, and vmPFC results shown in fig 1d support this claim, it wasn't clear to me that any of the results shown in any of the figures support the claim about MFG/SFG (dlPFC). Please revise the Discussion to match the Results (or vice versa).

Also, the results differ across the CeMe vs BLA roi's (cf. fig 1). This distinction should be clearly communicated here in the Discussion, not lumped.

Response: We apologize for the confusion here. The preference in the encoding of the CS+ and CS- for each region was determined based on the representational similarity analysis (RSA, Supplementary Fig. 7), which tested the similarity of regional contributions to the decoding in each time-block. For MFG and SFG, from Figure Sx we can see that they were clustered into different clusters across time-blocks. For example, MFG was clustered to the cluster that coded CS+ (the cluster on the bottom of each panel) at the first time-block of conditioning, but then clustered to the cluster that coded CS- (the cluster on the top of each panel) at the second time-block of conditioning. We now cited Supplementary Fig. 7 and clarified that the results were based on the RSA analysis following the statement. We also explicitly state that sub-nuclei of the amygdala showed different CS coding patterns. The revised sentences are as follows.

“... amongst others (results based on the representational similarity analysis, Fig. 3A and Supplementary Fig. 7). For example, sub-nuclei of the amygdala (BLA and CMA) exhibited preferences to CS+ during early extinction learning, while BLA exhibited preferences to CS- and CMA exhibited preferences to CS+ types during early threat conditioning and extinction memory recall (Fig. 1D). This observation is consistent with rodent and human studies showing that amygdala contributes to threat and safety processing in an anatomically specific way^{30,63–65}.”

Comment 20: *"We refer to those neural nodes as 'flexible' coders. Why do they change their encoding of the CS type?"*

I may have missed it, but the authors do not seem to formally test whether "they change" and it is not clear which of the color patches shown in fig 1d are actually significant. In some cases (e.g. CeMe, vmPFC), visual inspection suggests something more like habituation then "changing" or "flipping." Consider the vmPFC sphere. Muted red at the onset of acquisition (significant?) ...then "nothing" ... then dark-blue at the onset of extinction ... then muted blue for the remainder. As it stands, I'm not sure whether this is "even" interpreting the null. Please revise the Discussion to match the Results (or vice versa).

Response: This point is well taken. But even if it is habituation, it is still a change. We now modify the Results and Discussion sections to reflect what the reviewer is suggesting:

“It is important to note that we did not conduct significance tests on the changes of encoding across trial-blocks within each of these subregions, and that these changes may reflect those similar to a habituation effect.” (Results section)

“One possible explanation for this change is that it may reflect training-induced dynamic change in neural representation- similar to a habituation effect.” (Discussion section)

Comment 21: *The authors do not really address the larger significance of the decoders. Now that they exist and have been validated, how might they be applied. Note: I strongly encourage the authors to share their models with the scientific community via Neurovault, OSF, Github, or another public repository. This maximizes their value and increases the likelihood that they will be further developed and refined.*

Response: We totally agree with the reviewer. We now share our codes, masks and the trained models on GitHub: <https://github.com/zhenfu-wen01/threat-mvpa>. As for the applications, we added the following to the Discussion section of the manuscript:

“A potential application for the decoders is in neuromodulation studies where we could envision inducing some neuromodulation on one or more of the nodes from the extended

circuit and evaluate how such manipulation might change the decoding accuracies and the dynamic interactions and encodings we observed.”

Comment 22: *Authors need to very briefly acknowledge limitations of the study (e.g. the gap between predictive multivoxel models and causal insights licensed by mechanistic studies); these could even be framed as key challenges for future research.*

Response: We added a paragraph to acknowledge limitations of the study as below.

“Several limitations of this study should be noted. First, the masks for some ‘threat circuit’ regions, including vmPFC, sgACC, and dACC, were defined using small spheres around coordinates derived by meta-analysis. These masks could not fully cover the three relatively large cingulate cortices. Thus, patterns of activations within voxels outside of these masks may have contributed to the ‘whole-brain’ classification. We note that the size of the mask definition is unlikely to have impacted the ‘whole-brain’ classification results. This is because we conducted additional analyses using much larger masks (thus excluding more voxels from the same cingulate regions) and the outcome of the ‘whole-brain’ classifiers did not change (Supplementary Fig. 8). Second, the regional coding preference pattern summarized in Fig. 3A (consistent for CS+, CS-, or flexible) represents a general schematic illustration of the activation patterns specific to our experimental design and its structure. We speculate that while the engagement of these brain regions across different experimental stimuli and experimental design might be consistent, their decoding patterns across experimental phases might differ from one study to another. Further studies are needed to examine the dynamic neural representations of these regions across paradigms. Lastly, the results obtained from this study should not be misinterpreted to infer causality, given the indirect nature of fMRI data and the lack of direct manipulations of the brain signals. Future mechanistic studies are needed to test how the patterns of activations we observed might influence, or lead to, behavioral changes pertaining to the expression and extinction of threat responses across experimental phases.”

6. FIGURES.

Comment 23: *- figure 1 ... it is not intuitively clear that the columns in panel d indicate bins. This should be clearly labeled.*

Response: We modified Figure 1 accordingly.

For Brain Imaging Papers

Comment 24: *Is the length of each trial and interval between trials specified? Useful to provide Pavlovian cue duration(s).*

Response: We added this information to the updated manuscript to add more details of the paradigm.

“The trial structure was similar across experimental phases. Specifically, each trial started by presenting a context image (either a library or an office) for 3 seconds, after which the CS was presented for 6 seconds. In a reinforced CS+ trial, a shock was delivered at the end of the CS for 500 milliseconds. In other CS trials, no shock was delivered. A fixation was then presented during the inter-trial intervals, which ranged between 12 and 18 seconds, with an average of 15 seconds. The order of trials was pseudorandom. The colored lights used as CS+s and CS- were counterbalanced across participants. The conditioning occurred in one context (e.g., the ‘office’ image) while the extinction learning

and extinction memory recall phases occurred in the other context (e.g., the 'library' image). The context assignments were counterbalanced across participants."

Comment 25: *Is the field strength (in Tesla), pulse sequence type (gradient/spin echo, EPI/spiral), field-of-view, matrix size, slice thickness, and TE/TR/ flip angle clearly stated? useful to provide at least some of these details in tabular form for the studies used for training and testing ... can skip field strength if all 3T; can skip sequence type and FOV; can report voxel size; can skip TE and flip angle; but should report TR.*

Response: We added two tables to summarize this information as listed in our response above.

Comment 26: *If there was data normalization/standardization to a specific space template, is the type of transformation (linear vs. nonlinear) used and image types being transformed clearly described?*

Not clear; it's not sufficient to just say fmripred defaults. both the normalization and T2-T1 co-reg should be described; the specific MNI atlas/template should be noted.*

Response: We added the following sentences to provide more details for the preprocessing steps.

"Imaging data were preprocessed using fMRIPrep 20.0.2⁷⁰. The T1-weighted (T1w) were corrected for intensity non-uniformity with N4BiasFieldCorrection (ANTs 2.3.3) and used as T1w-reference throughout the preprocessing. The T1w-reference was skull-stripped, segmented into cerebrospinal fluid, white-matter and gray-matter, and then spatially normalized into the Montreal Neurological Institute (MNI) space (MNI152NLin2009cAsym) through nonlinear registration with antsRegistration (ANTs 2.3.3). The functional images were head-motion corrected using mcflirt (FSL) and slice-timing corrected using 3dTshift (AFNI). The preprocessed functional images were then co-registered to the T1w-reference using flirt (FSL) with the boundary-based registration (nine degrees of freedom), and spatially normalized into the MNI152NLin2009cAsym space by applying the parameters obtained from T1w-reference spatial normalization. Normalized functional images were resampled to 2 × 2 × 2 mm voxel size using Lanczos interpolation (ANTs 2.3.3) and smoothed with a 6-mm full-width half-maximum (FWHM) Gaussian kernel."

MINOR / SPECIFIC

Comment 27: *the References are a bit of a mess; please carefully copy edit for consistency.*

Response: We apologize for the inconsistency; we carefully revised the references in the updated manuscript.

Comment 28: *"Past research within this field has primarily focused on the roles of a few subcortical and cortical structures in how the association between the conditioned and the unconditioned stimuli is formed, and how the defensive responses are generated and subsequently extinguished."*

This is misleading, at least in terms of the human imaging lit; 'intensively' focused or similar would be better aligned w/ the state of the science.

Response: We replaced 'primarily' with 'intensively'.

Comment 29: *"This approach has led to the notion of the so-called 'threat circuit' that mainly includes subregions of the amygdala, periaqueductal gray, hippocampus, medial prefrontal cortex, and insular cortex^{3,7-9}."*

would consider adding Shackman & Fox AJP 2021 &/or Fullana's fear signature meta-analysis.

also - given that the authors focus their analyses on the MCC/dACC, it might make sense to replace 'medial pfc' with 'cingulate' or similar.

Response: We thank the reviewer for the suggestions. We added the two references into the revised manuscript. Since the vmPFC is also included in the 'threat circuit', we prefer to use medial prefrontal cortex here.

Comment 30: *"This approach has led to the notion of the so-called 'threat circuit' that mainly includes subregions of the amygdala, periaqueductal gray, hippocampus, medial prefrontal cortex, and insular cortex^{3,7-9}. This concept suggests the functional specificity of these regions during the acquisition and extinction of conditioned threat^{3,4}. For example, the amygdala is a key node in the acquisition and extinction of conditioned threat; and the hippocampus is critical for contextual information processing."*

*Hard to follow the narrative thread, as written; the first sentence is clear and true to the mechanistic and imaging literatures...but it is not 100% clear what's meant by "suggests" ... do the authors mean that the existence of a 'threat-anticipation circuit' concept tempts scientists to over-interpret, and conceptualize these regions as *specific* to fear/anx (a reverse inference type fallacy? my confusion is amplified by the "for example" sentence, which seems focused on mechanistic evidence of causal necessity. please revise this section for clarity.*

Response: We modified these sentences for clarity. We now state:

"Across species, but especially in the rodent literature, data suggest some specific associations between localized functional activations of these nodes and particular behavioral expressions or processes during the acquisition and extinction of conditioned threat^{3,11,12}. For example, the amygdala is important for the expression of conditioned freezing responses and for extinction learning¹³; while the hippocampus is involved in contextual information processing⁴."

Comment 31: *"Although largely supported by cross-species research,"*

This seems potentially misleading/disingenuous. as the authors well know, the Fullana meta-analysis failed to reveal consistent/significant amygdala activation in human FCing studies; while the authors did demonstrate early amygdala activation in their recent PNAS report, this evidence is not specific to Cing, given the nature of the contingencies at the outset of the acquisition scanning.

Now, I recognize that this is the brief introduction to an empirical report -- not a comprehensive review -- so there's no need to get into all of the weeds, but this lack of consistency across species/methods needs to be acknowledged in passing. for instance, the authors could modify the clause to indicate.

"Although largely supported by cross-species research Ref1 (but see Ref2),"

Response: We reorganized the Introduction section and deleted "Although largely supported by cross-species research".

Comment 32: *"guide deliberative defensive actions"*

"we investigated the contribution of distributed neural systems in the formation of memory representations at higher cortical levels, which may be related to encoding deliberative defensive actions."

Do you really mean to say 'deliberative'? This seems unduly speculative for the introduction.

Response: We deleted the sentence: "which may be related to encoding deliberative defensive actions" in the revised manuscript.

Comment 33: *"While the 'threat circuit' is highly necessary for the detection and response to threat"*

This seems misleading, insofar as not every node in that network appears to be mechanistically critical.

Response: We modified the sentence as follows by using a less strong statement.

"While the 'threat circuit' **plays an important role** in the detection and response to threat"

Comment 34: *"To understand the specific value of the 'threat circuit' to the detection of threat, we first employed MVPA to examine neural patterns within this circuit in distinguishing stimuli that have been conditioned to signal threat from those that do not." In the absence of behavioral evidence of "detection" this seems misleading; please revise to better align with the strengths and limitations of the data and approach.*

Response: We agree with the reviewer. As mentioned in our response above, in the revised Introduction section the sentence is as below.

"In the first approach, we examined fMRI-based neural patterns within the 'threat circuit' in decoding stimuli that have been conditioned to signal threat from those that have been conditioned to signal safety."

Comment 35: *"In the second analytic approach, based on recent conceptualizations that more distributed neural systems are engaged in threat processing^{10,11,13–17}," "We tested recent conceptualizations that more distributed neural systems are engaged in threat processing^{10,13,14,25,67}."*

"Our results strongly support recent conceptualizations calling for the integration of broad networks required for the implicit and explicit processing of threat."

Authors may consider citing this lovely new paper - up to you.

Review Neurosci Biobehav Rev. 2023. The Nature and Neurobiology of Fear and Anxiety: State of the Science and Opportunities for Accelerating Discovery

Also - why are the citations slightly different across the 2 quoted sentences?

Response: We agree with the reviewer that the timely review paper well-matches the topic here. We added the paper into our reference. We also revised the citations to make them consistent across the manuscript.

Comment 36: *"1465"  1,465; same for "1000".*

Response: Corrected.

Comment 37: *"Participants. We analyzed data from a total of 1465 participants across multiple studies. Participants from the discovery dataset (N = 425) were combined from multiple studies^{18,20,55,56} and the data from the external datasets 1 (N = 98) and 2 (N = 127) were from two other studies^{29,30}."*

Authors should provide a table with basic demographic and scanning information for these studies

e.g. citation, N, % F, mean age, summary of diagnostic status as relevant, basic fMRI details, and whether each study was used for training/discovery or hold-out testing

Response: This information was included in the newly added table.

Comment 38: *minor - it would be useful to know the duration (or range of durations) for the Pavlovian cues.*

Response: We added the information to the Methods section. The duration of each Pavlovian cue was 6 seconds.

Comment 39: *very minor - rationale for using $y = -21$ for splitting anterior from posterior hipp should be provided.*

Response: This decision was made based on a previous study (Poppenk et al. 2013). We added the reference into the Methods section.

Comment 40: *"We then transformed the model weights as suggested in a previous study³¹ to obtain the predictive weight of each voxel."*

Authors might explicitly term this "Haufe-transformed" as in this recent report from Avram Holmes & BTT Yeo.

Response: We revised the sentence accordingly:

"We then transformed the model weights **using the Haufe transformation** to obtain the predictive weight of each voxel³¹."

Comment 41: *"Validation on external datasets using different paradigms. We tested the sensitivity and specificity of the newly established circuit in identifying conditioned threats on external datasets 3-9."*

Hard to follow a written; by which i mean that Luke's IAPS dataset - for instance - does not include conditioned threats; please revise for clarity.

Response: We revised the sentence as follows.

Classifiers trained with brain activations of the extended circuit were tested on external datasets 3-9, which including threat conditioning paradigms, paradigm assessing 'subjective fear' to frightening images, as well as paradigms probing negative affects using intrinsically salient stimuli."

Comment 42: *"To compare the decoding accuracies across different paradigm categories, we divided the external datasets 3-9 into 3 groups: conditioned threat (datasets 3-6), 'subjective fear' (dataset 7), and emotionally salient stimuli (datasets 8-9)." Pavlovian threat cues are emotionally salient; better to say something like 'naturalistic'.*

Response: Thank you for the suggestion. We replaced 'emotionally salient stimuli' with 'intrinsically salient stimuli'.

Comment 43: *"The 'threat circuit'-based decoding models were successfully generalized to the two external datasets (Figures 1B, 1C)."*

Authors should summarize the results in the main text, as they did in the earlier section.

Response: We agree with the reviewer that summarizing the generalization results would be more informative. But adding all these numbers may decrease the readability of the main text. We added a Supplementary table to list all the validation results, and point the readers to the Supplementary table for details as follows.

"The 'threat circuit'-based decoding models were successfully generalized to the two external datasets (Figures 1B, C, **see Supplementary Table 2 for details**)."

Supplementary Table 2. Classification performance on validation datasets 1-2.

Phase	'Threat circuit' patterns		Whole-brain patterns	
	Validation dataset 2	Validation dataset 2	Validation dataset 1	Validation dataset 2
Conditioning TB1	58.30%	67%***	80.2%***	87.4%***
Conditioning TB2	64.6%**	82.7%***	70%***	90.6%***
Conditioning TB3	70.8%***	72.4%***	81.3%***	82.7%***
Conditioning TB4	70.8%***	71.7%***	77.1%***	77.2%***
Extinction TB1	68.1%***	58.70%	73.4%***	60.3%*
Extinction TB2	54.30%	50.80%	64.9%**	60.3%*
Extinction TB3	51.10%	50.00%	54.30%	57.90%
Extinction TB4	50.00%	48.40%	50.00%	55.60%
Recall TB1 (CS+E vs. CS-)	69.0%***	71.8%***	66.7%**	71.8%***
Recall TB1(CS+U vs. CS-)	66.7%**	71%***	71.3%***	80.6%***
Recall TB2 (CS+E vs. CS-)	56.30%	51.60%	58.60%	58.90%
Recall TB2 (CS+U vs. CS-)	46.00%	55.60%	59.80%	58.10%

Accuracies significantly higher than change level (two-sided binomial test, $p < 0.05$) were bolded. TB: Trial-block; ***: $p < 0.001$; **: $p < 0.01$; *: $p < 0.05$.

Comment 44: "stayed even" use formal English please; please polish

- "The improved classification performance of the 'whole-brain' pattern over the 'threat circuit' stayed even when the number of voxels was matched (Figure S2)."
"improved"  numerically superior

[note - i may have missed it, but i do not think the authors formally tested the difference]

Response: We replaced 'stayed even' with 'remained', and 'improved' with 'numerically superior' according to the reviewer's comment. The revised sentence is as follows:

"The **numerically superior** classification performance of the 'whole brain' pattern over the 'threat circuit' **remained** when the number of voxels was matched (Supplementary Fig. 3)."

Comment 45: "We used representational similarity analysis³³ to test the contribution of each node within this updated circuit to the decoding of threat and safety (Supplemental Figure 2)."

Typo? wrong fig cited?

Response: We apologize for the typo; we fixed it by citing the correct figure (Supplemental Fig. 7) in the revised manuscript.

Comment 46: "Lastly, we investigated how reliable and specific";

"More importantly...in a reliable and reproducible manner—classifiers based on activation patterns from these nodes can be generalized to different conditioning paradigms."; "provides a cohesive integrative extended threat circuit that encodes for threat and safety, consistently, specifically, and reliably";

"reliable" has a specific meaning in the psychometrics lit; please avoid using it here; these analyses gauge convergent/divergent validity ("specificity") not "reliability".

Response: We revised the manuscript according to the reviewer's comments by avoiding the use of 'reliable' or 'reliability'.

Comment 47: *"we investigated how reliable and specific the extended circuits are when classifying threat-conditioned stimuli across different paradigms"*

Isn't it singular "circuit"? or do you mean circuit(s)/model(s) because they are trained separately on each bin? please revise for clarity and consistency. perhaps makes sense to use a term such as "extended threat circuit decoders" or "binwise threat circuit decoders"

Response: We appreciate the reviewer's suggestion. We revised the sentence as follows. *"we investigated how sensitive and specific the extended circuit decoders are when classifying threat-conditioned stimuli across different paradigms."*

Comment 48: *"Some of these datasets included variant Pavlovian conditioning paradigms, and others included negative affect paradigms."*

Drift in nomenclature; sometimes described as 3 sets of dataset; here described as 2, ignoring the "subjective fear" distinction.

Response: We revised the language across the manuscript so that the descriptions are consistent.

"These datasets included variant paradigms of Pavlovian conditioning, paradigm assessing 'subjective fear' to frightening images, and paradigms of negative affect to intrinsically salient stimuli."

Comment 49: *"We first tested the optimized models" extended models.*

Response: We corrected it accordingly.

Comment 50: *"visual cues but assessed 'subjective fear',"*

Revise to make it clear that these are not mere 'visual cues' but are photos of ostensibly frightening animals/scenes

Response: We revised the sentence as follows.

"We next applied the models to another dataset (N = 65) that assessed 'subjective fear' induced by frightening images,"

Comment 51: *"emotionally salient stimuli"*

Pavlovian threat cues are emo'ly salient ... this should revise to make it clear that there are intrinsically aversive stimuli.

Response: We used 'intrinsically salient stimuli' instead.

Comment 52: *"not related to associative learning and memory." this is potentially Confusing as written; of course, Ss learn about and form memories of these cues...just say something about unlearned/intrinsically aversive.*

Response: We revised the sentence accordingly.

Comment 53: *figure s2 - are the x-axis labels artifactually compressed?*

Response: Thank you for pointing this out. We revised the figure accordingly.

Comment 54: *figure s3 - text labels (mni coordinates) are too small to read; please revise.*

Response: We revised the figure.

Comment 55: *"The accuracies of the classifiers were reduced when decoding neural representations about picture-evoked negative affect²⁸"*

"reduced"  "numerically lower (but still significantly above chance)"

Response: We revised the text accordingly.

Comment 56: not clear why the authors do not show the grey "zone" corresponding to chance performance in figure 3b-3i; also the colors used for the box-and-whiskers is not consistent with the earlier figures where, for example, green = extinction recall ... please revise for consistency

Response: We added the chance level areas to the figure based on the reviewer's suggestion. The four box plots in Figure 3B-3I are based on models from the four trial-blocks of conditioning phase (all were previously marked as blue). We used four different colors (randomly selected) here to make the box plots easier to distinguish. We do realize that using 'green' here is a little confusing, thus we changed the color scheme of these panels so that no color here is same as earlier figures. The updated figure is as follows.

Figure 3B-I. Figure legends are in the main text.

Comment 57: "most of the models showed reduced accuracies for physical pain" Most of the decoders showed numerically reduced accuracies for physical pain

Response: We revised the text accordingly.

Comment 58: *"Decoding accuracies for conditioned threat (datasets 3-6) were comparable to those for 'subjective fear' (dataset 7) (Chi-square test, TB1-TB4: all $\chi^2 < 5.2$, $p > 0.08$), and significantly higher than those for emotionally salient stimuli (datasets 8-9) at trial-blocks 2-4 (TB1-TB4: $\chi^2 = 1.1$, $p = 0.31$; $\chi^2 = 47.5$, $p < 0.001$; $\chi^2 = 30.6$, $p < 0.001$; $\chi^2 = 55.6$, $p < 0.001$). Overall, these results support the specificity of the decoding models to the neural correlates of stimuli associated with the conditioned threat." No they do not - Becker's fear photos work as well as Milad's conditioned visual threat cues; please revise to align with the actual results.*

Response: We revised the sentence as follows.

"Overall, these results suggest that conditioned threat and 'subjective fear' exhibited overlapped neural representations, which are different from those evoked by intrinsically salient stimuli."

Comment 59: *"The trained classifiers are sensitive and reproducible regardless of experimental paradigms or fMRI scanners."*

You could go a step further and mention across countries ... data are collected in different continents!

Response: Thank you for the suggestion. We revised the sentence as follows.

"The trained classifiers are sensitive and reproducible regardless of experimental paradigms or MRI scanners across countries."

Comment 60: *"Our results continue to highlight the critical role of the 'threat circuit' in the detection and response to threat.";*

"In addition to the 'threat circuit', our data highlight the value of other critical nodes that are also important in detecting and processing threat";

"More importantly, we demonstrated that these neural nodes contribute to the detection and processing of threat and safety cues in a reliable and reproducible";

These are observational/recording, not mechanistic/perturbation data; please don't use the term 'critical' ... likewise, these results speak to "classification" not detection or orchestrating defensive responses.

Response: We revised these sentences accordingly.

Comment 61: *"Importantly, our results advance current conceptions in the field by showing that sensory and cognitive neural nodes are also essential in enhancing the classification accuracies related to threat and safety processing."*

How does numerically increased classification accuracy advance or refine theory? authors need to unpack the claim.

Response: We refer to the reviewer to their comment below and the reference to the quote from our manuscript. The numerically increased classification accuracy in advancing our concepts and theories by showing the value of other nodes (other than the commonly considered 'core nodes'/usual suspects). These are really wonderful and great suggestions that are adding so much to the interpretation of our results- we greatly appreciate it. For the referred sentence here, we modified it as follows:

"Importantly, our results support current conceptions in the field by showing that sensory and cognitive neural nodes are also essential in enhancing the classification accuracies related to threat and safety processing; thereby highlighting the importance of considering

multiple neural systems in concert for a comprehensive understanding of the underlying mechanism of threat encounters.”

Comment 62: *"In addition to the 'threat circuit', our data highlight the value of other critical nodes that are also important in detecting and processing threat. When we encounter threat, we need to process all sensory modalities currently present, pay attention to our surroundings, retrieve prior memories related to the encounter of this or similar threats, and decide how to respond. Indeed, our results do show a critical engagement of multiple sensory and cognitive systems during the encounter of threat and safety cues, which is likely needed for the processing of these complex representations. These regions spread across the primary sensory and motor cortex, frontal-parietal cortex, and cerebellum."*

it is perhaps worth noting that the present results provide compelling new empirical evidence for theories that make exactly this point -- cf. "Question 3. Afterword" in Grogans et al... Review Neurosci Biobehav Rev. 2023: The Nature and Neurobiology of Fear and Anxiety: State of the Science and Opportunities for Accelerating Discovery

Response: We totally agree with the reviewer. We added the following sentence to make it clear.

"Our results provide compelling empirical evidence for conceptual frameworks that threat processing involves multiple processes encompassing multiple brain circuits^{14,21}."

Comment 63: *"and the dorsal anterior cingulate cortex (dACC)"*

Why is this acronym redefined in the discussion; please fix.

Response: We removed the acronym redefinition in the Discussion.

Comment 64: *"Our analyses of voxel contributions to the decoding of CS+ and CS- across experimental phases revealed interesting predictive patterns (summarized in Figure 3a)." More appropriate to cite figs 1d *and* 3a*

Response: We revised the text by citing Fig. 1D and 3A.

Comment 65: *"These brain regions continued to signal, or encode, the CS+ even during extinction learning and during extinction memory retrieval, consistent with maintaining some representation of the initial conditioned threat memory regardless of extinction. These data provide neurobiological support from the human brain to the now well-accepted concept that extinction does not erase the original conditioned threat associations, first proposed by Pavlov in 1927."*

This section needs some nuance. In fact, fig 1a reminds us that decoder accuracy drops to chance levels for most of extinction (bins 2-4) and recall with extinction (bin 2). Please revise to better convey this nuance.

[Aside - As noted above, the extinction and recall data are real strengths of the paper, and I think these results are cool as-is.]

Response: Thank you for this note. As noted earlier in response to another comment, extinction learning happens really quick in humans- 1-2 trials usually. We are observing during extinction learning and also during extinction recall some patterns of activations that are consistent with encoding threat-related processes. And thus, during extinction learning, but as the reviewer points out, during extinction recall, we still see patterns resembling those noted during threat encoding- suggesting that during extinction recall, we now see patterns related to extinction and patterns related to the original association. We made some adjustments to the text in the Introduction and Discussion sections to

further enhance this point. For example, we added the following sentences to the Discussion section:

“The diminished difference between CS+ and CS- led to rapidly decreased decoding accuracy after early extinction which ended up in chance-level performances in late extinction (Fig. 1A-C, 2A-C). The same pattern of drop in accuracies was also observed during extinction memory recall. This is the case for the extinguished (CS+E) and unextinguished (CS+U) cues. The rapid decrease in accuracies is consistent with the well-documented exceptionally quick extinction learning in humans both during extinction learning and extinction memory recall^{3,66}.”

Comment 66: *“Despite all of these variations, the models trained on one paradigm were still highly sensitive and specific in distinguishing threatening from non-threatening conditioned cues across paradigms, suggesting that the classifiers learned some general patterns of conditioned threat.”*

Please cite fig 3b-3d.

Response: We fixed it accordingly.

Comment 67: *“were applied to studies that used auditory fear conditioning, supporting the idea that” only 1 study, not studies.*

Please cite fig 3e

Response: We fixed it accordingly.

Comment 68: *“And as expected, the accuracies of our models tested in the negative affect and pain processing studies were substantially lower compared to those obtained from the threat conditioning studies.”*

Significantly lower;

Cite fig 3g-3i;

Consider mentioning the chi-square tests (which support the claim of 'significantly lower') i.e. “Decoding accuracies for conditioned threat (datasets 3-6) were comparable to those for ‘subjective fear’ (dataset 7) (Chi-square test, TB1-TB4: all $\chi^2 < 5.2$, $p > 0.08$), and significantly higher than those for emotionally salient stimuli (datasets 8-9) at trial-blocks 2-4 (TB1-TB4: $\chi^2 = 1.1$, $p = 0.31$; $\chi^2 = 47.5$, $217 < 0.001$; $\chi^2 = 30.6$, $p < 0.001$; $\chi^2 = 55.6$, $p < 0.001$). Overall, these results support the specificity of the decoding models to the neural correlates of stimuli associated with the conditioned threat.”

Response: We followed the reviewer’s suggestion and revised the sentence as follows.

“And as expected, the accuracies of our models tested in the negative affect and pain processing studies (Fig. 3G-I) were significantly lower (based on Chi-square test) compared to those obtained from the threat conditioning studies (Fig. 3B-E).”

Comment 69: *“In summary, our results contribute to the expansion of the ‘threat circuit’ by including multiple other sensory and cognitive neural nodes.”*

Do they? authors need to carefully explain the value of mvpa classifiers for inference in the introduction, and they need to be more precise here, in the discussion, about what these results mean. if the goal is accurate decoding of pavlovian conditioned threat, you need to include regions well beyond ‘the usual suspects’ ... but those ‘extended’ regions are also less likely to be specific to ‘threat’ ... as it stands, the degree to which ‘traditional’ vs. ‘extended’ regions show specificity across tasks is not addressed by the present results, so more care needs to be taken in how the results are framed.

Response: Of course we agree. The extended network we discuss are important contributors to the perception, detection, processing, and expression. Sensory processing nodes included in our circuit are in way meant to say that this is the only function this structure does for a living. We are highlighting the importance and need to include other nodes to 'the usual suspects' for enhanced accuracies for our models so that we can have a more comprehensive representation of threat processing and its extinction- we need sensory and cognitive systems to work with the usual suspect to get the job done. We feel that this is an important point that this reviewer is raising across multiple points, and we thoroughly revised the Discussion section to address this clearly across the manuscript. Specifically for this sentence, we revised it as follows:

"In summary, **our results point to the important contribution of multiple sensory and cognitive neural nodes to the decoding of stimuli associated with threat detection and responding.**"

Comment 70: *"provides a cohesive integrative extended threat circuit that encodes for threat and safety, consistently, specifically, and reliably"*

This claim goes well beyond the data; there is nothing about MVPA (regularized multiple logistic regression) that guarantees "cohesive/integrative" ... it's blind statistical prediction of cue-type ... please fix

Response: The sentence was meant to relay the idea that added information or processing is now gained with this methodological approach. We revised the sentence to make it a bit softer here:

"...our study **provides evidence of an integrative circuit** that encodes for threat and safety..."

Comment 71: *Authors are strongly encouraged to add an Author Contributions section to the paper or supplement.*

Response: We added an Author Contributions section in the updated manuscript as below.

“Author contributions

Z.W., E.A.P., J.E.L., and M.R.M. conceived and designed the study. E.F.P.-S., S.W.L., J.R., and F.A. collected the data. Z.W. analyzed the data under the supervision of M.R.M.. Z.W. and M.R.M. interpreted the data with input from all authors. Z.W. and M.R.M. drafted the manuscript. All authors revised the manuscript and approved its final version for submission.”

Reviewer #3

General comment: *Wen and colleagues investigated distributed brain signals linked to conditioned threat in the human brain. This is an excellent study that provides a clear contribution to the field. It addresses a timely and important question concerning the existence of a spatially distributed, “extended threat” circuit. The paper is well written and the methods appear sound. I have only a few points of clarification that I outline next:*

Response: We are immensely grateful for the positive feedback, and sincerely appreciate the insightful comments and suggestions. We have made a sincere effort to carefully address your requests, suggestions, and concerns point-to-point as detailed below.

Comment 1: *To define the extended circuit, the authors adopted a series of criteria that, while reasonable, were ad hoc. The authors state that 7 blocks with “robust” classification were used. How was “robust” defined?*

Response: We apologize for not describing the criteria clearly. We considered ‘robust’ classification as significant decoding accuracies in both discovery dataset and validation datasets 1-2 (Fig. 2A-C). All 4 time-blocks of conditioning, the first 2 time-blocks of extinction learning, the first time-block of extinction learning recall (both CS+E vs. CS- and CS+U vs. CS-) met this criterion. We did not include the second time-block of extinction learning to the analysis, because the accuracy dropped a lot from the first to the second time-block (both in the discovery dataset, and in the validation dataset 1), and the accuracies in the second time-block for the two validation datasets were lower than 65% (64.9% and 60.3%). We clarified this in the revised manuscript.

“... across the 7 trial-blocks that showed robust classification performances. across the 7 time-blocks that showed robust classification performances⁷⁸. We considered ‘robust’ classification as significant decoding accuracies in both discovery dataset and validation datasets 1-2 (Fig. 2A-C). All 4 time-blocks of conditioning, the first 2 time-blocks of extinction learning, and the first time-block of extinction learning recall (both CS+E vs. CS- and CS+U vs. CS-) met this criterion. We did not include the second time-block of extinction learning in the analysis, because the accuracy dropped a lot from the first to the second time-block (both in the discovery dataset, and in the validation dataset 1), and the accuracies in the second time-block for the two validation datasets were lower than 65% (64.9% and 60.3%).”

Comment 2: *Then, 10% of voxels with the largest contributions were used. While a more thorough analysis of how these choices impact the results is perhaps beyond the scope of the paper, it would be important to provide some sense of the overall stability of the results in terms of these choices. For example, if the extended circuit looks quite different with 20% of the voxels, what would that mean?*

Response: We appreciate the reviewer’s comment. The choice of 10% of voxels with the largest contributions was made because the mean percentage of voxels that significantly contributed to the classification ($p < 0.05$, FDR-corrected) across the trial-blocks (i.e., brain maps in Fig. 2D) was around 10% (9.5%). Our main purpose is to demonstrate that there are many other neural systems beyond the ‘threat circuit’ that are also engaged in threat conditioning and extinction, but not to thoroughly identify all the brain regions that are involved in the processing (also see our response to **Comment 3** below). Thus, these identified 14 regions are representative of neural systems involved in sensory and cognitive processes. To the reviewer’s concern about the overall stability of the results, we examined the brain maps with two different criteria (10%, and 20% as suggested by the reviewer). As shown in the figure below, the two criteria resulted in similar brain regions,

suggesting the robustness of the results in term of the choices. We added the following sentences to the Method section to clarify it.

“This choice was made because the mean percentage of voxels that significantly contributed to the classification ($p < 0.05$, FDR-corrected) across the trial-blocks (Fig. 2D and Supplementary Fig. 4) was around 10% ($9.5\% \pm 2.2\%$). Also, the results were robust overall regarding the choices of percentage (Supplementary Fig. 6).”

Supplementary Figure 6. Voxels with the largest contributions to the classification across trial-blocks. A. The top 10% voxels with the largest contributions. **B.** The top 20% voxels with the largest contributions. Regions included in the ‘extended circuit’ are marked.

Comment 3: *In this context, I feel torn about the inclusion of the OFC. The first time I noticed that it had been treated differently was in the Methods, but maybe I missed it before. I think the text of the Results and then the Discussion should discuss this more explicitly so that it’s not buried in the Methods (but I might have missed it).*

Response: We agree with the reviewer. We moved the following sentences into the main text now.

“Although the OFC was not among the regions that made the largest contributions, we included it because it highly contributed in the first trial-block of conditioning, which is consistent with literature suggesting its important role in threat-related processing^{38,39}.”

We also added some discussion points about the inclusion of the OFC in the Discussion section as follows:

“While the OFC only significantly contributed to the early conditioning phase, we included it because it was reported to be involved in threat-related processing across studies^{38,39}. There might be other regions that are only transiently engaged at specific stages of conditioning and/or extinction (like the OFC here), that were not identified. Our objective here is not to thoroughly identify all the brain regions that are involved in the processing. Rather, we are highlighting a more integrative view in aggregating the ‘threat circuit’ together with other neural systems involved in sensory and cognitive processes, to study neural mechanisms underlying threat encounter, as proposed in recent conceptualizations^{14,15,20–24}.”

Comment 4: *I’m having some trouble connecting the extended circuit so nicely summarize in Fig 3A to the results in Fig 2D. My suggestion to help the reader better appreciate the relationship would be to have more transparent brains in 2D with the outlines of the regions*

shown in 3A on them. In any case, I think it's important to help the reader get a better appreciation of the contributions of the extended circuit in the actual brain maps.

Response: We appreciate the reviewer's thoughtful suggestion. We now marked the regions included in the 'extended circuit' on the brain maps as below (Supplementary Fig. 5). Since this figure is too large to combine with Fig. 2A-C in one page, and it is not very compact, we kept this figure as a supplementary figure to Fig. 2D. And as shown in our response to the reviewer's comment above, we added a brain map that showed the 10% most contributed voxels across trial-blocks to help readers get a better sense of the contributions of the 'extended circuit'. We also shared all these brain maps together with the brain masks of the 'extended circuit' in GitHub for readers that are interested in them.

Minor:

Comment 5: *ROIs: hopefully the authors will share their ROI masks so that others can also employ them, but an additional sentence or two should indicate more clearly how the CMA/BLA masks were defined.*

Response: We shared our codes, ROI masks, and trained models on GitHub now: <https://github.com/zhenfu-wen01/threat-mvpa>.

Supplementary Figure 5. Predictive patterns of voxels across the brain. A voxel with red/blue color indicates that this voxel is more activated to CS+/CS-. Permutation tests were conducted to assess the voxel contributions to the classification. Only voxels that

significantly contributed to the classification ($P < 0.05$, FDR-corrected) are shown. Regions included in the 'extended circuit' were marked.

Reviewer #1 (Remarks to the Author):

1. Thank you for the response, I appreciate the additional analysis. However, I don't think that it addresses the removal of the signal from the previous trial. I understand that the classifiers were trained on the beta maps (my "time points" phrasing perhaps wasn't the clearest, I apologize). The concern is that the preceding shock signal is picked up by the classifier (i.e. not the outcome from the current trial). A simple analysis would be to show that the results are the same for cases where the previous trial was a shock (or other aversive stimulus) versus where it was not a shock. Modeling GLM using autoregressive models is known to suffer from poor removal of signal (please refer to the paper that I shared above), therefore, I would not find modeling shock events sufficient. That being said, I agree that the lack of aversive events during extinction speaks towards a true "threat" signal rather than a residual.

Overall, this is the authors' own work and I would leave the final choice on them. I think that this paper has the potential to become an important piece of the literature, and eliminating all doubt would significantly strengthen it, so my suggestion would be to include the analysis based on preceding outcome.

The remainder of the comments fully addressed the posed questions. I particularly thank the authors for sharing the code and maps online.

Reviewer #2 (Remarks to the Author):

Revision of 'Distributed neural representations of conditioned threat in the human brain'
Wen...& Milad
#####

Despite its basic and translational importance, the neurobiology of fear remains incompletely understood, impeding the development of more effective interventions for maladaptive fear. Most human imaging studies employ small samples and conventional mass univariate analyses, which have important inferential limitations. Thus, the focus of this paper is timely and likely to be of interest to a broad spectrum of researchers and clinical scientists.

Key Strengths

- extraordinarily large sample for this domain of research, with the potential for enhanced power, precision, and reproducibility
- fairly comprehensive tests of sensitivity, specificity, and generalizability via multiple independent hold-out datasets (ie. not used for classifier training)
- the innovative MVPA classifier approach has the potential to address questions of specificity that cannot be addressed using traditional mass univariate approaches, innovation is strengthened by the thoughtful bin-wise approach, which provides important clues about temporal dynamics
- the first author (Dr. Wen) is to be congratulated on what amounts to a truly enormous amount of work (e.g. data wrangling, processing, and analyses for 9 datasets)
- code shared via Github!

Comments on the Revised MS

I congratulate the authors' on a highly responsive set revisions which, in general, are thorough and thoughtful. The revised paper is much stronger and more useful to the affective neuroscience community.

Despite this enthusiasm, I remain somewhat concerned about the Juelich BLA ROI alignment.

Taking care to use the same template used by the authors (mni_icbm152_t1_tal_nlin_asym_09c.nii), I visually compared the figure that they kindly added to the Supplement against the underlying T1 anatomy, the original 2005 Amunts ROI-validation paper, and the 4th edition of the Mai atlas. By eye, it looks like the BLA masks are centered on the temporal horn of the lateral ventricle, below the amygdala.

Reading through the authors' response to R3, I then realized that they kindly made the masks available on Github (fearNet_10regions_MNI152NLin2009cAsym_res-2_space.nii.gz regions 1 and 2). Inspection of the binarized masks (especially in more caudal regions) reinforces my concern that the BLA ROI includes a sizable number of voxels outside of BLA proper, including adjacent ventricle, WM, and cortical GM.

Unless I am misunderstanding what the authors are doing, which is certainly possible, this should be corrected.

Reviewer #3 (Remarks to the Author):

The authors have adequately addressed my comments and suggestions.

Reviewer #1 (Remarks to the Author):

Comment 1: *Thank you for the response, I appreciate the additional analysis. However, I don't think that it addresses the removal of the signal from the previous trial. I understand that the classifiers were trained on the beta maps (my "time points" phrasing perhaps wasn't the clearest, I apologize). The concern is that the preceding shock signal is picked up by the classifier (i.e. not the outcome from the current trial). A simple analysis would be to show that the results are the same for cases where the previous trial was a shock (or other aversive stimulus) versus where it was not a shock. Modeling GLM using autoregressive models is known to suffer from poor removal of signal (please refer to the paper that I shared above), therefore, I would not find modeling shock events sufficient. That being said, I agree that the lack of aversive events during extinction speaks towards a true "threat" signal rather than a residual.*

Overall, this is the authors' own work and I would leave the final choice on them. I think that this paper has the potential to become an important piece of the literature, and eliminating all doubt would significantly strengthen it, so my suggestion would be to include the analysis based on preceding outcome.

Response: We apologize for misunderstanding the reviewer's comment in our previous revision and sincerely appreciate the suggestion. To verify that the preceding shock signal was not the main contributor to the classification performance, we conducted an additional analysis as proposed. Specifically, for the classification analysis, we exclusively selected CS+ trials (and corresponding CS- trials) that were not preceded by the delivery of a shock and also that there is no delivery of a shock within this trial. This gives us the purest signal where there is no influence by a shock within the CS or preceding the CS, as recommended by the reviewer. Based on this analysis, we obtained similar classification performance (shown below in Supplementary Fig. 3) as in the main analysis, thus mitigating the main concern regarding the contribution of the shock signal on the classification, and supporting our findings from the classification results from early extinction learning and extinction recall as the reviewer noted. We added the following sentences to the revised manuscript, and added the figure to the Supplementary material.

“Considering that the autoregressive model of GLM suffers from removing temporal autocorrelation (Mumford et al. 2012), the preceding shock signal might be picked up by the classifier. To investigate this issue, we conducted an additional analysis exclusively

using trials not preceded by a shock and not paired with a shock. The results (Supplementary Fig. 3) demonstrate that the classification performance was not predominantly contributed by shock signal.”

Reference: Mumford, J. A., Turner, B. O., Ashby, F. G. & Poldrack, R. A. Deconvolving BOLD activation in event-related designs for multivoxel pattern classification analyses. *NeuroImage* 59, 2636–2643 (2012).

Supplementary Figure 3. Classification in the conditioning phase by only including trials not confounded by shock. Classification performance based on 'threat circuit' patterns on the discovery dataset (A), validation dataset 1 (B), and validation dataset 2 (C). Classification performance based on 'whole-brain' patterns on the discovery dataset (D), validation dataset 1 (E), and validation dataset 2 (F). To fully exclude the impact of shock on the classification, we exclusively included trials that are not followed by a shock, and not preceded by a shock (so that the preceding shock signal is not picked up by the

classifier) for the classification. This resulted in 2 CS+ trials and 2 CS- trials in each trial-block for the classification analysis.

Comment 2: The remainder of the comments fully addressed the posed questions. I particularly thank the authors for sharing the code and maps online.

Response: We thank the reviewer for the positive comments.

Reviewer #2 (Remarks to the Author):

General comment: *Despite its basic and translational importance, the neurobiology of fear remains incompletely understood, impeding the development of more effective interventions for maladaptive fear. Most human imaging studies employ small samples and conventional mass univariate analyses, which have important inferential limitations. Thus, the focus of this paper is timely and likely to be of interest to a broad spectrum of researchers and clinical scientists.*

Key Strengths

- *extraordinarily large sample for this domain of research, with the potential for enhanced power, precision, and reproducibility*

- *fairly comprehensive tests of sensitivity, specificity, and generalizability via multiple independent hold-out datasets (ie. not used for classifier training)*

- *the innovative MVPA classifier approach has the potential to address questions of specificity that cannot be addressed using traditional mass univariate approaches, innovation is strengthened by the thoughtful bin-wise approach, which provides important clues about temporal dynamics*

- *the first author (Dr. Wen) is to be congratulated on what amounts to a truly enormous amount of work (e.g. data wrangling, processing, and analyses for 9 datasets)*

- *code shared via Github!*

Response: We are very grateful for the reviewer's positive comments.

Comments on the Revised MS

Comment: *I congratulate the authors' on a highly responsive set revisions which, in general, are thorough and thoughtful. The revised paper is much stronger and more useful to the affective neuroscience community.*

Despite this enthusiasm, I remain somewhat concerned about the Juelich BLA ROI alignment. Taking care to use the same template used by the authors (mni_icbm152_t1_tal_nlin_asym_09c.nii), I visually compared the figure that they kindly added to the Supplement against the underlying T1 anatomy, the original 2005 Amunts

ROI-validation paper, and the 4th edition of the Mai atlas. By eye, it looks like the BLA masks are centered on the temporal horn of the lateral ventricle, below the amygdala.

Reading through the authors' response to R3, I then realized that they kindly made the masks available on Github (fearNet_10regions_MNI152NLin2009cAsym_res-2_space.nii.gz regions 1 and 2). Inspection of the binarized masks (especially in more caudal regions) reinforces my concern that the BLA ROI includes a sizable number of voxels outside of BLA proper, including adjacent ventricle, WM, and cortical GM.

Unless I am misunderstanding what the authors are doing, which is certainly possible, this should be corrected.

Response: We thank the reviewer for the thoughtful comments on the amygdala masks. Considering the reviewer's comments, we decided to redefine the CMA and BLA masks by using amygdala masks constructed by Shackman and colleagues, which provide enhanced anatomical sensitivity and selectivity compared to the Juelich/SPM amygdala masks. We re-conducted the analysis with the new masks, and obtained similar results as using the Juelich/SPM masks. We updated the figures accordingly (e.g., see Fig. 1 below), and revised the description of the mask definition in the manuscript as below.

"The CMA and BLA masks were defined based on amygdala masks constructed by Shackman and colleagues (Tillman et al. 2018). These masks provide enhanced anatomical sensitivity and selectivity compared to the Juelich/SPM amygdala masks (see Tillman et al. 2018). We added a coronal view of the amygdala masks (Supplementary Fig. 10) to enable readers to judge the alignment of the masks with the MNI152 template. Here, the CMA was defined by combining the central nucleus and medial nucleus. The BLA was defined by combining the lateral nucleus, basolateral and basomedial nuclei."

Reference: Tillman, R. M. et al. Intrinsic functional connectivity of the central extended amygdala. *Hum. Brain Mapp.* 39, 1291–1312 (2018).

Figure 1. Classification analyses based on neural activations of the ‘threat circuit’.

Reviewer #3 (Remarks to the Author):

Comment: The authors have adequately addressed my comments and suggestions.

Response: We thank the reviewer for the positive comments.

Reviewer #1 (Remarks to the Author):

Thank you for addressing the last concern. I have no further comments.

Reviewer #2 (Remarks to the Author):

Great work - I have no further suggestions.